# *Dlk1-Dio3* locus-derived lncRNAs perpetuate postmitotic motor neuron cell fate and subtype identity

Ya-Ping Yen[1,2], Wen-Fu Hsieh[3], Ya-Yin Tsai[1], Ya-Lin Lu[1], Ee Shan Liau[1], Ho-Chiang Hsu[1], Yen-Chung Chen[1], Ting-Chun Liu[1], Mien Chang[1], Joye Li[3], Shau-Ping Lin[2]*, Jui-Hung Hung[3,4]*, Jun-An Chen[1]*

[1]Institute of Molecular Biology, Academia Sinica, Taipei, Taiwan, Republic of China; [2]Institute of Biotechnology, College of Bio-Resources and Agriculture, National Taiwan University, Taipei, Taiwan, Republic of China; [3]Institute of Bioinformatics and Systems Biology, National Chiao Tung University, Hsinchu, Taiwan, Republic of China; [4]Department of Computer Science, National Chiao Tung University, Hsinchu, Taiwan, Republic of China

*For correspondence:
shaupinglin@ntu.edu.tw (SL);
juihunghung@gmail.com (JH);
jachen@imb.sinica.edu.tw (JC)

**Competing interests:** The authors declare that no competing interests exist.

**Abstract** The mammalian imprinted *Dlk1-Dio3* locus produces multiple long non-coding RNAs (lncRNAs) from the maternally inherited allele, including *Meg3* (i.e., *Gtl2*) in the mammalian genome. Although this locus has well-characterized functions in stem cell and tumor contexts, its role during neural development is unknown. By profiling cell types at each stage of embryonic stem cell-derived motor neurons (ESC~MNs) that recapitulate spinal cord development, we uncovered that lncRNAs expressed from the *Dlk1-Dio3* locus are predominantly and gradually enriched in rostral motor neurons (MNs). Mechanistically, *Meg3* and other *Dlk1-Dio3* locus-derived lncRNAs facilitate Ezh2/Jarid2 interactions. Loss of these lncRNAs compromises the H3K27me3 landscape, leading to aberrant expression of progenitor and caudal *Hox* genes in postmitotic MNs. Our data thus illustrate that these lncRNAs in the *Dlk1-Dio3* locus, particularly *Meg3*, play a critical role in maintaining postmitotic MN cell fate by repressing progenitor genes and they shape MN subtype identity by regulating *Hox* genes.

## Introduction

Investigations of the gene regulatory networks involved in cell-type specification during embryonic development have been protein-centric for decades. However, given the prevalence of high-throughput sequencing analyses of mammalian genomes, it is now appreciated that non-coding RNAs (ncRNAs) account for at least 50~80% of transcriptomes (*Pauli et al., 2011*; *Rinn and Chang, 2012*). Regulatory ncRNAs can be broadly classified based on their size (*Mattick, 2009*). Short RNA species (~20–30 nucleotides [nt]), such as microRNAs (miRNAs), have emerged as pivotal modulators of development and disease through mediation of translational repression or mRNA degradation (*Esteller, 2011*). Long non-coding RNAs (lncRNAs; >200 nt) are gaining prominence for their roles in many cellular processes, from chromatin organization to gene expression regulation during embryonic development (*Kung et al., 2013*; *Rinn and Chang, 2012*; *Rutenberg-Schoenberg et al., 2016*). Thus, it is not surprising that lncRNAs were recently found to be associated with an array of diseases including cancers, as well as cardiovascular and neurological disorders (*Briggs et al., 2015*).

Accumulating evidence supports that lncRNAs can induce *cis-* and *trans-*acting gene silencing. For example, the lncRNA *Airn* directly represses the paternally-expressed *Igf2r* gene *in cis* for the maintenance of ESC differentiation (*Pauler et al., 2005*) and *Xist* lncRNA triggers *in cis* inactivation of the X chromosome (*Lee, 2009*). The human lncRNA *HOTAIR*, which is expressed from the caudal

**eLife digest** When a gene is active, its DNA sequence is 'transcribed' to form a molecule of RNA. Many of these RNAs act as templates for making proteins. But for some genes, the protein molecules are not their final destinations. Their RNA molecules instead help to control gene activity, which can alter the behaviour or the identity of a cell. For example, experiments performed in individual cells suggest that so-called long non-coding RNAs (or lncRNAs for short) guide how stem cells develop into different types of mature cells. However, it is not clear whether lncRNAs play the same critical role in embryos.

Yen et al. used embryonic stem cells to model how motor neurons develop in the spinal cord of mouse embryos. This revealed that motor neurons produce large amounts of a specific group of lncRNAs, particularly one called *Meg3*. Further experiments showed that motor neurons in mouse embryos that lack *Meg3* do not correctly silence a set of genes called the *Hox* genes, which are crucial for laying out the body plans of many different animal embryos. These neurons also incorrectly continue to express genes that are normally active in an early phase of the stem-like cells that make motor neurons.

There is wide interest in how lncRNAs help to regulate embryonic development. With this new knowledge of how *Meg3* regulates the activity of *Hox* genes in motor neurons, research could now be directed toward investigating whether lncRNAs help other tissues to develop in a similar way.

*HOXC* locus, acts *in trans* to target the *HOXD* cluster for gene silencing (*Li et al., 2013*; *Rinn et al., 2007*). Approximately 20% of lncRNAs are associated with polycomb repressive complex 2 (PRC2) (*Zhao et al., 2010*; *Khalil et al., 2009*), which is comprised of many subunits and functions to deposit histone H3K27 trimethylation (H3K27me3) and to suppress gene expression (*Di Croce and Helin, 2013*; *Margueron and Reinberg, 2011*; *Simon and Kingston, 2013*). Although some evidence indicates that lncRNAs might serve as scaffolds for PRC2 assembly and guide PRC2 to specific genomic targets, whether the interaction is specific and necessary in development or disease contexts is still unclear (*Cifuentes-Rojas et al., 2014*; *da Rocha et al., 2014*; *Davidovich and Cech, 2015*; *Davidovich et al., 2015*; *Davidovich et al., 2013*; *Kaneko et al., 2014a*; *Kaneko et al., 2014b*). Therefore, it is imperative to demonstrate that the specific interactions of lncRNAs with the PRC2 complex are functionally important and have specific regulatory targets to direct development or induce disease in vivo.

We used spinal motor neuron (MN) differentiation as a paradigm to assess these interactions. Although spinal cord development is one of the best characterized processes in the central nervous system (CNS) (*Alaynick et al., 2011*; *Catela et al., 2016*; *Chen et al., 2011*; *Mazzoni et al., 2013a*; *Mazzoni et al., 2013b*; *Narendra et al., 2015*; *Philippidou and Dasen, 2013*), how lncRNAs are involved in its transcription factor-driven gene regulatory networks is unclear (*Briscoe and Small, 2015*). MN differentiation into subtypes is mediated by the mutually exclusive expression of Hox transcription factors, which is programmed according to the body segment along the rostrocaudal (RC) axis. For example, segmental identity of MNs is defined by the mutually exclusive expression of Hox6, Hox9 and Hox10 (*Dasen et al., 2003*; *Lacombe et al., 2013*). In each segment, MNs are grouped into different columns according to their innervating targets. For instance, within the brachial Hox6$^{on}$ segment, MNs are further grouped into axial muscle projecting MNs (Lhx3$^{on}$, MMC) and forelimb-innervating MNs (Foxp1$^{on}$, LMC). Finally, another set of mutually exclusive Hox proteins, such as Hox5 and Hox8 expression in the Foxp1$^{on}$ LMC, further controls the rostral and caudal motor pool identity, which directs motor pools to either innervate proximal or distal muscles in the forelimb (*Catela et al., 2016*; *Dasen et al., 2005*).

In the spinal cord, polycomb proteins control the exclusion of certain Hox protein expression at specific RC positions and maintain this repression in differentiated cells. Depletion of the polycomb repressive complex 1 (PRC1) component Bmi1 at brachial level causes ectopic expression of Hoxc9 and subjects LMC neurons to a thoracic preganglionic column (PGC) fate. Conversely, elevation of Bmi1 represses Hoxc9 at thoracic level and subjects PGC neurons to an LMC fate (*Golden and Dasen, 2012*). These observations suggest that specific Hox repression may be maintained in MNs by distinct PRC1 activity levels, programmed along the RC axis. Recently, it was shown that during

MN differentiation, *Hox* chromatin is demarcated into discrete domains controlled by opposing RC patterning signals (i.e., retinoic acid (RA), Wnt, and fibroblast growth factors (FGFs)) that trigger rapid and domain-wide clearance of H3K27me3 modifications deposited by PRC2 (*Mazzoni et al., 2013b*). More specifically, RA activates retinoic acid receptors (RARs) and binds to the *Hox1~5* chromatin domains, which is followed by synchronous domain-wide removal of H3K27me3 to acquire cervical spinal identity. At the tailbud, a gradient of Wnt and FGF signals induces expression of the Cdx2 transcription factor that binds and clears H3K27me3 from the *Hox1~Hox9* chromatin domains, thereby establishing brachial or thoracic segmental identity (*Mazzoni et al., 2013b*). Together, these findings indicate that epigenetic regulation of *Hox* clusters is critical to initiate and maintain patterns of *Hox* expression and that cross-repressive interactions of combinations of Hox proteins later consolidate the diversification of postmitotic MNs. However, the underlying mechanism that demarcates the histone modifiers at a molecular level is still unclear. Although many lncRNAs are known to regulate these histone modifiers, whether lncRNAs are directly involved in MN fate determination remains to be established.

We found that lncRNAs in the imprinted *Dlk1-Dio3* locus are highly enriched in postmitotic MNs. The *Dlk1-Dio3* locus contains three protein-coding genes (*Dlk1*, *Rtl1*, and *Dio3*) from the paternally inherited allele, and multiple lncRNAs and small ncRNAs are derived from the maternally inherited allele, including *Meg3*, *Rian* (containing 22 box C/D snoRNAs), as well as the largest miRNA megacluster in mammals (*anti-Rtl1*, which contains the *miR-127/miR-136* cluster of 7 miRNAs, and *Mirg* that within the *miR-379/miR-410* cluster). Interestingly, all of the ncRNAs are regulated by a common *cis*-element and epigenetic control, resulting in a presumable large polycistronic transcription unit (*Das et al., 2015*; *Lin et al., 2003*; *Seitz et al., 2004*). Although the *Dlk1-Dio3* locus is well known to play crucial roles in stem cells (*Lin et al., 2007*; *Lin et al., 2003*; *Qian et al., 2016*), we unexpectedly found that expressions of *Meg3* and other lncRNAs from the *Dlk1-Dio3* locus are also all enriched in postmitotic MNs. However, whether this locus functions during neural development had not been explored previously. Here, we show that lncRNAs in the imprinted *Dlk1-Dio3* locus shape postmitotic MNs by inhibiting progenitor and non-neural genes, and they also control MN subtype identity by regulating Hox expression. Our results provide strong evidence for the critical function of lncRNAs during MN development, emphasizing their physiological functions during embryonic development.

## Results

### Identification of cell-type-specific lncRNAs during MN differentiation

As epigenetic landscape remodelling and the cell fate transition during MN differentiation are well characterized (*Chen et al., 2011*; *Li et al., 2017*; *Tung et al., 2015*), we took advantage of an ESC differentiation approach that can recapitulate MN development to systematically identify cell lncRNAs during this differentiation process. Firstly, an ESC line harbouring the MN transgenic reporter *Hb9::GFP* was harnessed into MNs (*Wichterle et al., 2002*), and we then sequentially collected RA-induced nascent neural epithelia (Hoxa1[on], NE at day 2), MN progenitors (Olig2[on], pMN at day 4), and postmitotic MNs (Hb9::GFP[on], postmitotic MNs at day 7) by fluorescence-activated cell sorting (FACS). Simultaneously, spinal interneurons (INs) derived from [smoothened agonist; SAG][low] conditions were collected and Hb9::GFP[off] cells were sorted at day 7 as controls (*Figure 1A*). Next, we performed strand-specific RNA-seq across libraries preserving non-polyadenylated transcripts while removing ribosomal RNAs, since many lncRNAs are non-polyadenylated (*Yin et al., 2012*; *Zhang et al., 2012*), and carried out de novo transcriptome assembly (*Qian et al., 2016*) to discover novel lncRNAs that might be specifically enriched during MN development (detailed in the Materials and methods and summarized in *Figure 1—figure supplement 1A*; *Supplementary file 1*). Several known markers for each cell type during MN differentiation were accurately recovered, corroborating the high quality and specificity of our RNA-seq data (*Figure 1B*). Our approach yielded 10,177 lncRNAs, 752 of which (7.39%) were previously unidentified from the Ensemble mm10 database. We also found that 4295 (77.78%) of our identified lncRNAs overlapped with recently reported spinal MN-related lncRNAs, which were discovered by poly A[+]-enriched RNA-seq approaches (*Amin et al., 2015*; *Narendra et al., 2015*) (*Figure 1—figure supplement 1B*). Finally, we removed minimally expressed transcripts (TMM normalized read

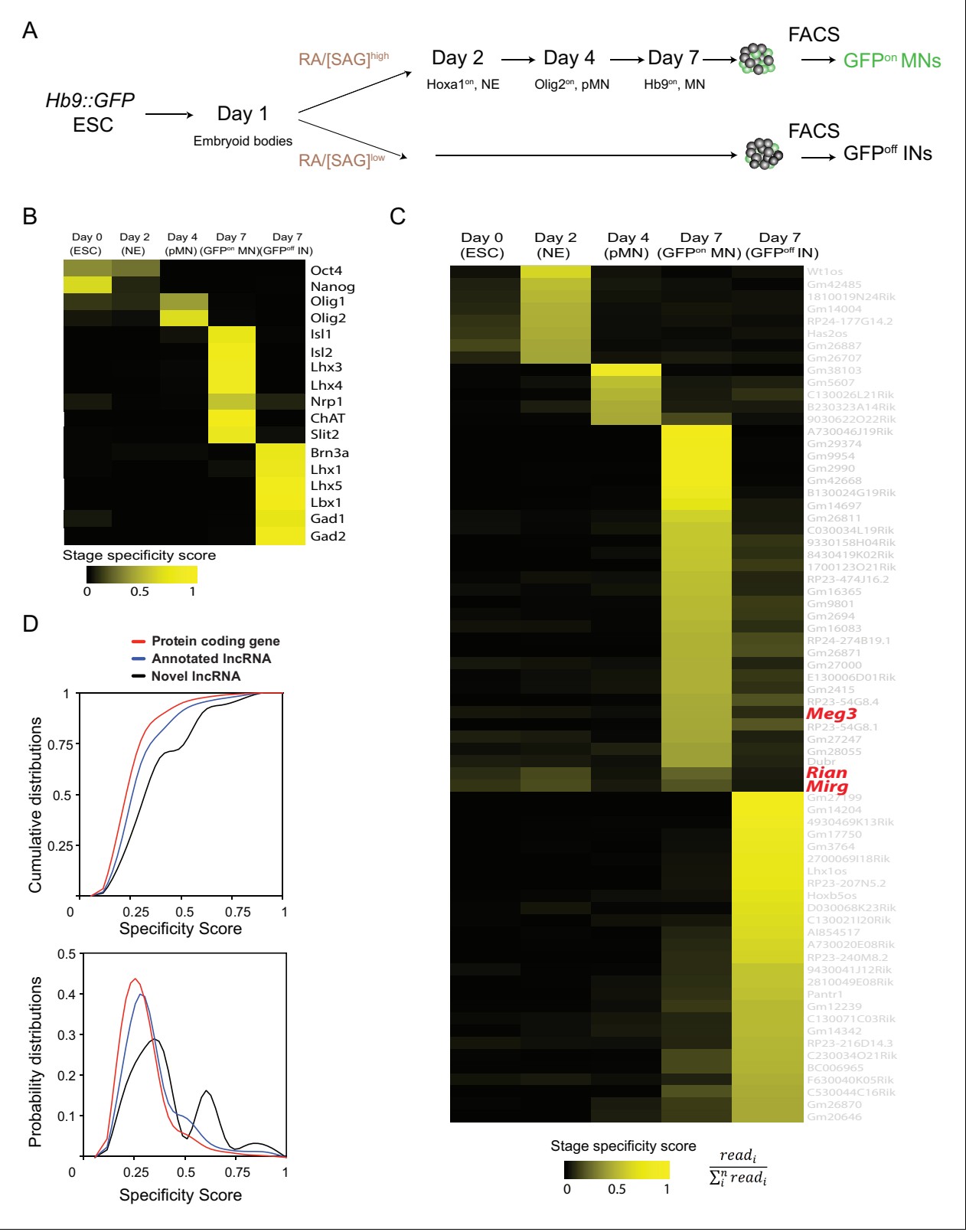

**Figure 1.** Identification of cell-type-specific lncRNAs during motor neuron differentiation. (**A**) Schematic illustration of the differentiation process from *Hb9::GFP* ESCs to spinal MNs. RA: retinoic acid. SAG: Smoothened agonist. ESC: embryonic stem cell. NE: neural epithelium. pMN: motor neuron progenitor. MN: motor neuron. IN: interneuron. (**B and C**) Heatmaps presenting the abundances of known cell transcription factors (**B**) and the abundances of lncRNA signatures (**C**) across each stage from ESCs to postmitotic MNs and INs (color indicates specificity scores). (**D**) Cumulative

*Figure 1 continued on next page*

*Figure 1 continued*

distributions (above) and probability distributions (below) of the stage specificity score of different categories of genes (protein coding genes [red], annotated lncRNAs [blue] and novel lncRNAs [black]), representing a measure of differential expression for each transcript across the cell types. The distribution reveals that annotated lncRNAs and novel lncRNAs manifest significantly higher specificity (according to Kolmogorov-Smirnov tests) than protein-coding genes.

The online version of this article includes the following figure supplement(s) for figure 1:

**Figure supplement 1.** Systematic identification and verification of motor neuron signature lncRNAs.

count <10 in all samples), which left 602 expressed lncRNAs (*Supplementary file 2*). Based on stage-specific scores (see Materials and methods), 70 stage-signature lncRNAs during the ESC~MNs differentiation process were uncovered (ESC, NE, pMN, MN, and IN in *Figure 1C*). Compared to protein-coding genes, both annotated lncRNAs and novel lncRNAs (newly identified in our de novo transcriptome assembly) had higher cell-type specificity (*Figure 1D*, Kolmogorov-Smirnov test, p=1.41 $\times$ 10$^{-9}$ and p=3.53 $\times$ 10$^{-7}$, respectively; see Materials and methods), implying that lncRNAs might play specific roles in each cell type during ESC~MN differentiation.

## The *Dlk1-Dio3* locus-derived lncRNAs are enriched in the nuclei of postmitotic MNs

To identify developmentally up-regulated lncRNAs, we compared day 4 pMNs vs. day 7 postmitotic MNs (*Figure 2A*). Furthermore, to retrieve cell-type-specific lncRNAs, we performed a pairwise comparison of day 7 postmitotic MNs against day 7 INs (*Figure 2B*). We identified 117 lncRNA candidates from our analysis as being postmitotic MN-enriched. We further selected several MN-lncRNAs that manifested high normalized reads from RNA-seq data and verified their MN-specific expression by qPCR (*Figure 1—figure supplement 1C*). Interestingly, the lncRNAs *Meg3*, *Rian*, and *Mirg*, which are transcribed from the imprinted *Dlk1-Dio3* locus on mouse chromosome 12qF, all manifested strong enrichment in postmitotic MNs (simplified schematic locus depicted in *Figure 2C*, detailed locus information in *Figure 2—figure supplement 1A*). Since these lncRNAs are conserved amongst placental mammals (*Ogata and Kagami, 2016*), we chose to characterize their functions in greater detail.

To investigate why *Meg3-Rian-Mirg* are highly enriched in postmitotic MNs, we examined the binding landscape of MN-specific transcription factors (i.e., Lhx3 and Isl1), histone modifications (H3K4me3 and H3K27ac), and chromatin accessibility (ATAC-seq) across the *Meg3-Rian-Mirg* locus from previously published studies (*Figure 2D*) (*Mazzoni et al., 2013a*; *Narendra et al., 2015*; *Rhee et al., 2016*). Within this locus, we uncovered an MN-specific active chromatin region that possesses enhancer/promoter characteristics with direct MN-specific transcription factor binding (*Figure 2D*). Furthermore, overexpression of MN-TFs in a maternally-inherited intergenic differentially methylated region deletion (*IG-DMR$^{mat\Delta}$*) ESC line, which leads to simultaneous silencing of all maternally-expressed lncRNAs in the *Meg3-Rian-Mirg* locus but leaves the MN-TF binding site intact (*Figure 2—figure supplement 1A*) (*Lin et al., 2007*; *Lin et al., 2003*), can robustly induce *Meg3* expression (*Figure 2—figure supplement 1B*). Therefore, we suggest that MN-TFs bind and directly activate *Meg3* during ESC~MN differentiation.

We further performed *Meg3* in situ hybridization and immunostaining of the adjacent sections along the RC axis from E10.5~12.5. We found that *Meg3* expression: (1) is enriched in the mantle zone of the developing spinal cord during development and is gradually enriched in postmitotic MNs (Isl1/2$^{on}$ cells) after E12.5; (2) has no preference for columnar MN subtypes, as revealed by Foxp1 (LMC-MNs) and Lhx3 (MMC-MNs) immunostaining; and (3) exhibits rostral high (brachial and thoracic) and caudal low (lumbar) asymmetry after E12.5 (*Figure 2E and F*).

Why does *Meg3* exhibit strong enrichment in the brachial spinal cord? Given that previous reports indicate that rostral *Hox* genes enriched in the brachial spinal cord are mediated by an RA gradient (*Mazzoni et al., 2013b*; *Novitch et al., 2003*), we hypothesized that *Meg3* might also be induced by RA. To examine this possibility, we checked if there is any RA-driven binding to RAR sites near the *Meg3* promoter (*Figure 2G*) (*Mahony et al., 2011*). Interestingly, we found that RA treatment results in novel binding of RAR directly to the *Meg3* promoter, as well as subsequent recruitment of the basal transcription complex (Pol2-S5P in *Figure 2H*). Moreover, we observed that *Meg3*

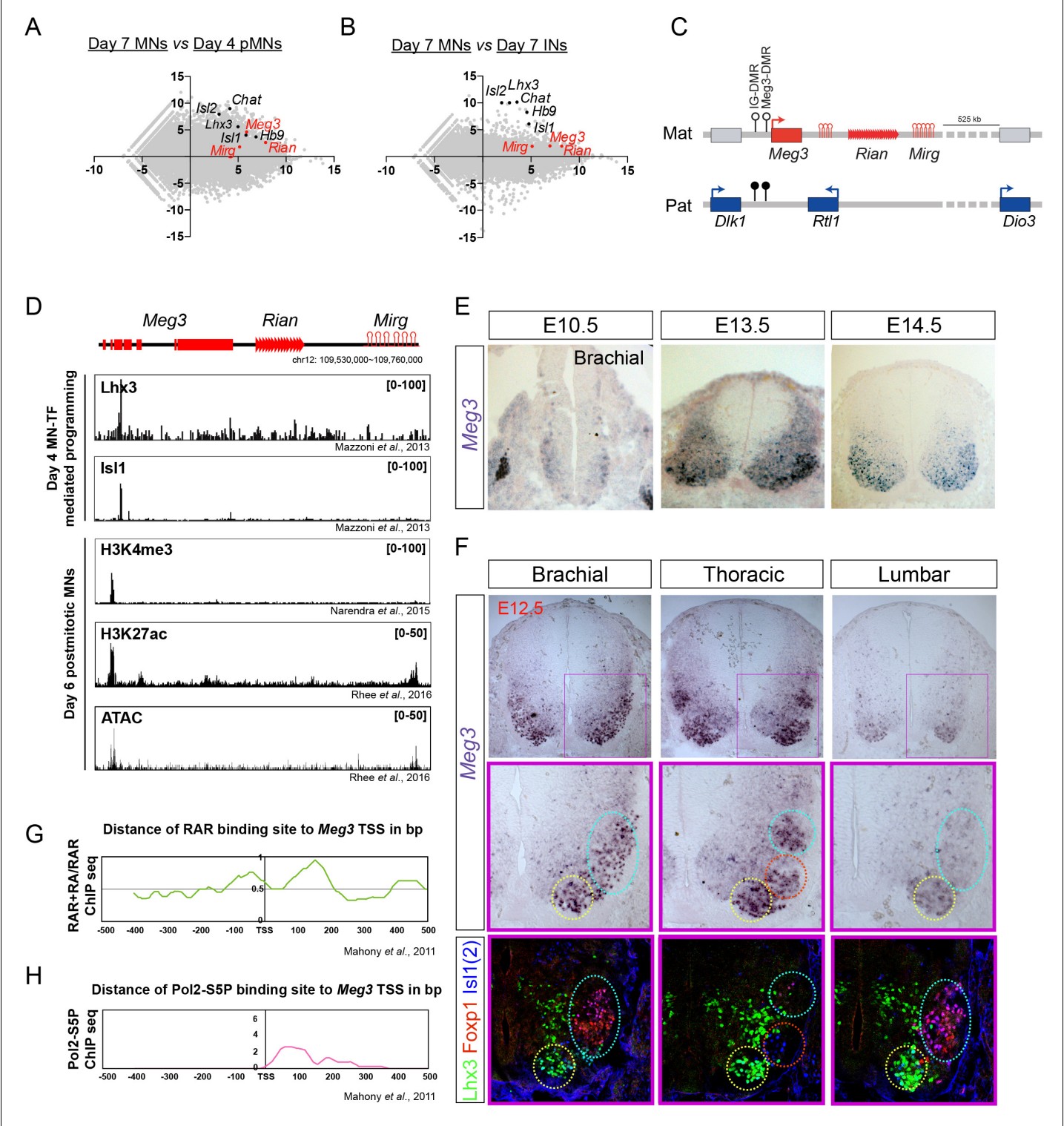

**Figure 2.** *Dlk1-Dio3* locus-derived lncRNAs are enriched in postmitotic motor neurons. (**A and B**) MA plots demonstrate that *Meg3*, *Rian*, and *Mirg* are postmitotic (day 7 MNs $_{vs.}$ day 4 pMNs; **A**) MN signature (day 7 MNs $_{vs.}$ day 7 INs; **B**) lncRNAs. *X*-axis: mean abundance; *Y*-axis: log$_2$ fold-change. (**C**) Illustration of the imprinted *Dlk1-Dio3* locus. The lncRNAs of the maternally-inherited allele (in red) are on mouse chromosome 12. The miRNA and C/D snoRNA genes are shown by hairpin loops and triangles, respectively. (**D**) Comparison of ChIP-seq for MN transcription factors (Lhx3 and Isl1), H3K4me3, and H3K27ac, together with ATAC-seq of the *Dlk1-Dio3* locus. (**E**) In situ hybridization shows that *Meg3* is gradually enriched and restricted in postmitotic MNs in the developing spinal cord. (**F**) In situ hybridization together with adjacent sections of immunostaining on E12.5 segmental spinal cords demonstrate that *Meg3* is enriched at brachial and thoracic MNs (Isl1/2$^{on}$), but no preference for columnar MN subtypes was revealed for Foxp1

*Figure 2 continued on next page*

*Figure 2 continued*

or Lhx3. (**G and H**) RAR binding and recruitment of the elongating form of Pol2 to the *Meg3* transcription start site (TSS) occur within 8 hr of retinoic acid (RA) exposure.

The online version of this article includes the following figure supplement(s) for figure 2:

**Figure supplement 1.** Characterization of the lncRNAs in the *Dlk1-Dio3* locus.

is induced after the addition of RA in *IG-DMR^matΔ^* ESCs after 8 hr (*Figure 2—figure supplement 1C*), indicating that RA/RAR activation triggers the strong *Meg3* expression in rostral brachial MNs.

Finally, to characterize the abundance and subcellular localization patterns of *Meg3* at a cellular level, we designed a set of single molecule RNA FISH probes specific to *Meg3* and examined their expression in ESC~MNs. We observed a speckled pattern of *Meg3* expression enriched in the nucleus, suggesting it has a potential function in gene regulation (*Figure 2—figure supplement 1D*). Furthermore, qPCR of subcellular-fractionated RNAs from ESC~MNs validated that *Meg3* is not only enriched in the nucleus, but that it is also chromatin-associated (*Figure 2—figure supplement 1E*; *Gapdh* as cytoplasmic marker, *Rnu1* (*U1* snRNA) as nuclear marker, and *Kcnq1ot1* as a chromatin-associated RNA control). Together, these findings suggest that lncRNAs in the *Dlk1-Dio3* locus are postmitotic MN-enriched, and that they are directly activated by MN-TFs and RA/RAR. At a cellular level, *Meg3* is highly enriched in MN nuclei and is chromatin-associated, indicating a potential function in chromatin regulation.

## *Meg3* facilitates interaction of the PRC2 complex with Jarid2 in ESC~MNs

While several previous reports have revealed that interactions between Jarid2 and ncRNAs regulate PRC2 recruitment to chromatin, including lncRNAs in the *Dlk1-Dio3* locus of ESCs (*da Rocha et al., 2014*; *Kaneko et al., 2014a*; *Kaneko et al., 2014b*), the roles of PRC2/Jarid2 in postmitotic cells are less clear. Unlike the PRC2 complex, Jarid2 is known to have diverse cell-type-specific functions (*Landeira and Fisher, 2011*). Surprisingly, we found that expression of *Jarid2* was reactivated in postmitotic MNs, pointing to a possible specific regulation in this cell type (*Figure 3—figure supplement 1A* upper panel) (*Takeuchi et al., 1995*). Moreover, compared to several known lncRNAs that interact with PRC2 complex, *Meg3* and *Rian* manifested much more abundant expressions in the postmitotic MNs (*Figure 3—figure supplement 1A* lower panel). This prompted us to examine if lncRNAs in the *Dlk1-Dio3* locus bind to the PRC2 complex and maintain postmitotic MN fate by controlling the H3K27me3 landscape. To test this hypothesis, we first demonstrated that immunoprecipitation (IP) of endogenous PRC2 complex components (i.e., Ezh2 and Suz12), as well as the PRC2 cofactor Jarid2, from ESC~MNs specifically retrieves *Meg3*, *Rian* and *Mirg* RNA, whereas the nuclear ncRNA *Rnu1* and the lncRNA *Malat1* were not captured by Ezh2, Jarid2, or Suz12 (*Figure 3A*). Given that *Rian* and *Mirg* are further processed to snoRNAs and miRNAs (*Lin et al., 2003*) and that *Meg3* is known to regulate pluripotency (*Stadtfeld et al., 2010*), imprinting (*Das et al., 2015*), and PRC2 function (*Zhao et al., 2010*), we focused on biochemical characterization of *Meg3*. Several *Meg3* isoforms have previously been documented, so we scrutinized across the entire *Meg3* locus (~31 kb) and found that two isoforms, *Meg3^v1^* and *Meg3^v5^* (Ensemble mm10 database), are predominantly expressed in Hb9::GFP^on^ MNs (*Figure 3—figure supplement 1B,C*). Moreover, *Meg3^v1^* and *Meg3^v5^* isoforms account for more than 99% of *Meg3* transcripts than other isoforms during ESC~MN differentiation. (*Figure 3—figure supplement 1D*) (*Kaneko et al., 2014b*; *Zhou et al., 2007*). Interestingly, *Meg3^v1^* and *Meg3^v5^* have mutually exclusive exon sequences (*Figure 3—figure supplement 1E*), raising the possibility that the two isoforms might exert different functions. However, both purified biotinylated *Meg3^v1^* and *Meg3^v5^* RNA retrieved Ezh2 from cell nuclear extracts of ESC~MNs (*Figure 3B*; *GFP* RNA was used as a negative control). These results suggest that these *Meg3* isoforms directly interact with PRC2/Jarid2 complexes and might facilitate association of the PRC2 complex with Jarid2.

As the PRC2 complex and Jarid2 are known to interact in a non-stoichiometric manner (*Pasini et al., 2010*; *Peng et al., 2009*), we further examined if *Meg3* facilitates the interaction between PRC2 complex and Jarid2. To test this possibility, we performed IP with Ezh2 (a core component of PRC2) to retrieve Jarid2 from ESC~MNs (*Figure 3C*). We first verified that *Meg3*

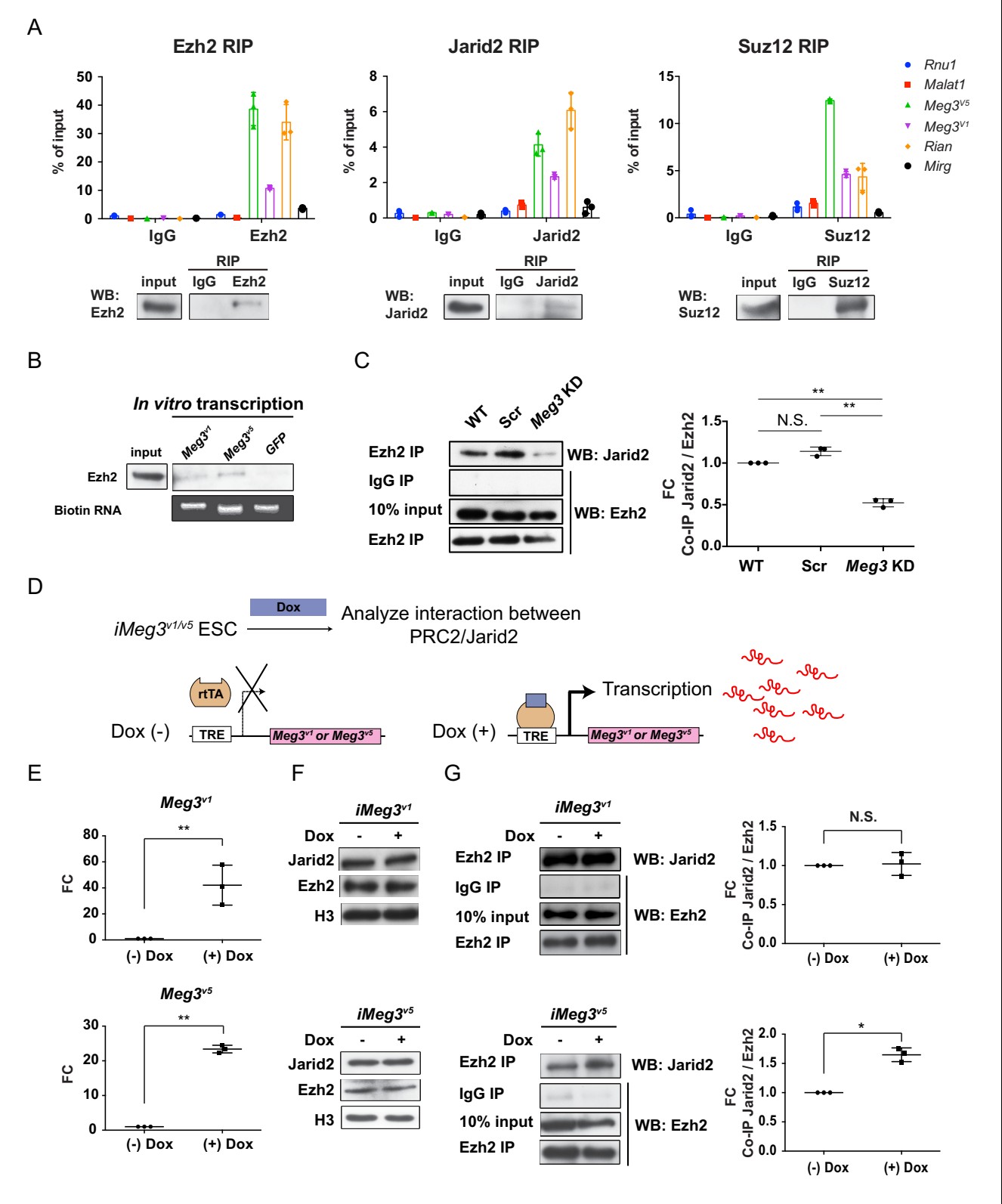

**Figure 3.** *Meg3* facilitates the non-stoichiometric interaction of the PRC2 complex and Jarid2. (**A**) Ezh2, Jarid2, and Suz12 immunoprecipitation specifically retrieves *Meg3* RNA isoforms (**v1 and v5**). *Rnu1* RNA and *Malat1* lncRNA are negative controls. 10% input was used to normalize the retrieval efficiency (error bars represent SD, n = 3 independent experiments). Immunoblotting reflects the recovery of Ezh2, Jarid2 and Suz12 proteins using the corresponding antibodies. (**B**) In vitro-transcribed, biotinylated *Meg3* RNA isoforms retrieved Ezh2. (**C**) Ezh2 interacts with Jarid2 in ESC~MNs,
*Figure 3 continued on next page*

Figure 3 continued

but knockdown of *Meg3* impairs this interaction. The abundance of Jarid2 is shown on the right (N.S.: not significant; error bars represent SD, n = 3 independent experiments; ** p-value<0.01 by Student's *t*-test). (D and E) The design of inducible 'Tet-On' ESC lines expressing *Meg3*$^{v1}$ or *Meg3*$^{5}$ under the doxycycline (Dox)-regulated promoter. In the presence of Dox, the reverse tetracycline-controlled transactivator (rtTA) is recruited to the TRE (tetracycline response element), thereby initiating robust transcription of *Meg3*$^{v1}$ or *Meg3*$^{v5}$, respectively. (F) Overexpression of *Meg3*$^{v1}$ or *Meg3*$^{v5}$ does not alter the protein levels of Ezh2 or Jarid2. (G) *Meg3*$^{v5}$ but not *Meg3*$^{v1}$ stimulates more Ezh2 and Jarid2 interaction. The abundances of Jarid2 are shown on the right (FC: fold-change; N.S.: not significant; error bars represent SD, n = 3 independent experiments; * p-value<0.05, ** p-value<0.01 by Student's *t*-test).

The online version of this article includes the following figure supplement(s) for figure 3:

**Figure supplement 1.** Characterization of *Meg3* isoforms.

knockdown (KD) did not affect the protein abundance of Ezh2/Jarid2, but we did observe that it undermined the interaction between Ezh2 and Jarid2, suggesting that *Meg3* facilitates this interaction (*Figure 3C* and *Figure 3—figure supplement 1F*). We then investigated if the two *Meg3* isoforms have differing abilities to facilitate Ezh2/Jarid2 binding by generating two locus-defined Tet-ON-inducible *Meg3* ESCs (*Figure 3D*, iMeg3$^{v1}$ and iMeg3$^{v5}$). Upon doxycycline induction, both *Meg3* isoforms were induced ~20–50 fold, yet the abundance of Ezh2/Jarid2 remained unaffected (*Figure 3E and F*). *Meg3*$^{v5}$ overexpression in ESC~MNs significantly increased the binding of Ezh2 and Jarid2, whereas *Meg3*$^{v1}$ overexpression *Figure 3E*; *Figure 3F* had a minimal effect (*Figure 3G*), indicating that *Meg3*$^{v5}$ is a strong facilitator of the binding of the PRC2 complex and Jarid2. Accordingly, we suggest that *Meg3*, and particularly the *Meg3*$^{v5}$ isoform, facilitates the binding of the PRC2 complex and Jarid2 in postmitotic MNs.

### *Dlk1-Dio3* locus-derived lncRNAs maintain the epigenetic landscape in postmitotic MNs

To test if the binding of Ezh2/Jarid2 by the lncRNAs in the *Dlk1-Dio3* locus is important to maintain the epigenetic landscape in postmitotic MNs, we systematically analyzed genome-wide H3K27me3 profiles of control and *IG-DMR*$^{mat\Delta}$ ESC~MNs by ChIP-seq (chromatin immunoprecipitation-sequencing) (*Figure 4A*). To overcome the complication of concomitant up-regulation of paternal coding genes in *IG-DMR*$^{mat\Delta}$ ESCs (*Lin et al., 2007*; *Lin et al., 2003*), we further established two retrovirus-based short hairpin RNAs (shRNAs) targeting *Meg3* and used a knockdown approach to prevent impairment of DMR sites. Both shRNAs reduced the expression of *Meg3* by an average of ~90% compared to endogenous levels in ESC~MNs (*Figure 4—figure supplement 1A*). As negative controls, we performed independent infections with retroviruses containing scrambled shRNA with no obvious cellular target RNA. We selected two stable *Meg3* KD ESCs (referred to as H6 and K4 hereafter) that had the best ESC morphology for further experiments. Verification by qPCR indicated that the expressions of two other lncRNAs, *Rian* and *Mirg*, from the maternal allele of the *Dlk1-Dio3* imprinted locus were all significantly down-regulated in *Meg3* KD MNs, whereas paternal genes were unaffected (*Figure 4—figure supplement 1A*). This finding is consistent with a previous report indicating that *Meg3-Rian-Mirg* probably represents a single continuous transcriptional unit (*Das et al., 2015*).

We then performed H3K27me3 ChIP-seq of control and *Meg3* KD MNs. We observed a trend of global down-regulation of H3K27me3 in both independent experiments of *Meg3* KD MNs, most likely a reflection of compromised Ezh2/Jarid2 interaction (*Figure 4—figure supplement 1B*). Since the response to PRC2 activity change in a given cell type might be context-dependent (*Davidovich et al., 2013*), we sought to identify relevant genes in MNs regulated by *Dlk1-Dio3* locus-derived lncRNAs based on the loss of H3K27me3. To achieve this, we profiled gene transcriptomes of control, *IG-DMR*$^{mat\Delta}$, and *Meg3* KD ESC~MNs. Next, we compared the co-upregulated genes between *IG-DMR*$^{mat\Delta}$ and *Meg3* KD ESC~MNs, together with H3K27me3 landscape upon the loss of ncRNAs in the *Dlk1-Dio3* locus (*Figure 4A* and *Figure 4—figure supplement 1C*). This approach revealed 585 genes in MNs that displayed down-regulation of the H3K27me3 epigenetic landscape and concomitant up-regulation of gene expression upon loss of the *Meg3* lncRNAs (*Figure 4A*). Gene ontology (GO) analysis of these genes revealed significant enrichment for RC patterning and progenitor genes, and strikingly so for homeodomain *Hox* genes (*Figure 4—figure supplement 1D*; false discovery rate (FDR) q-value ≤0.05).

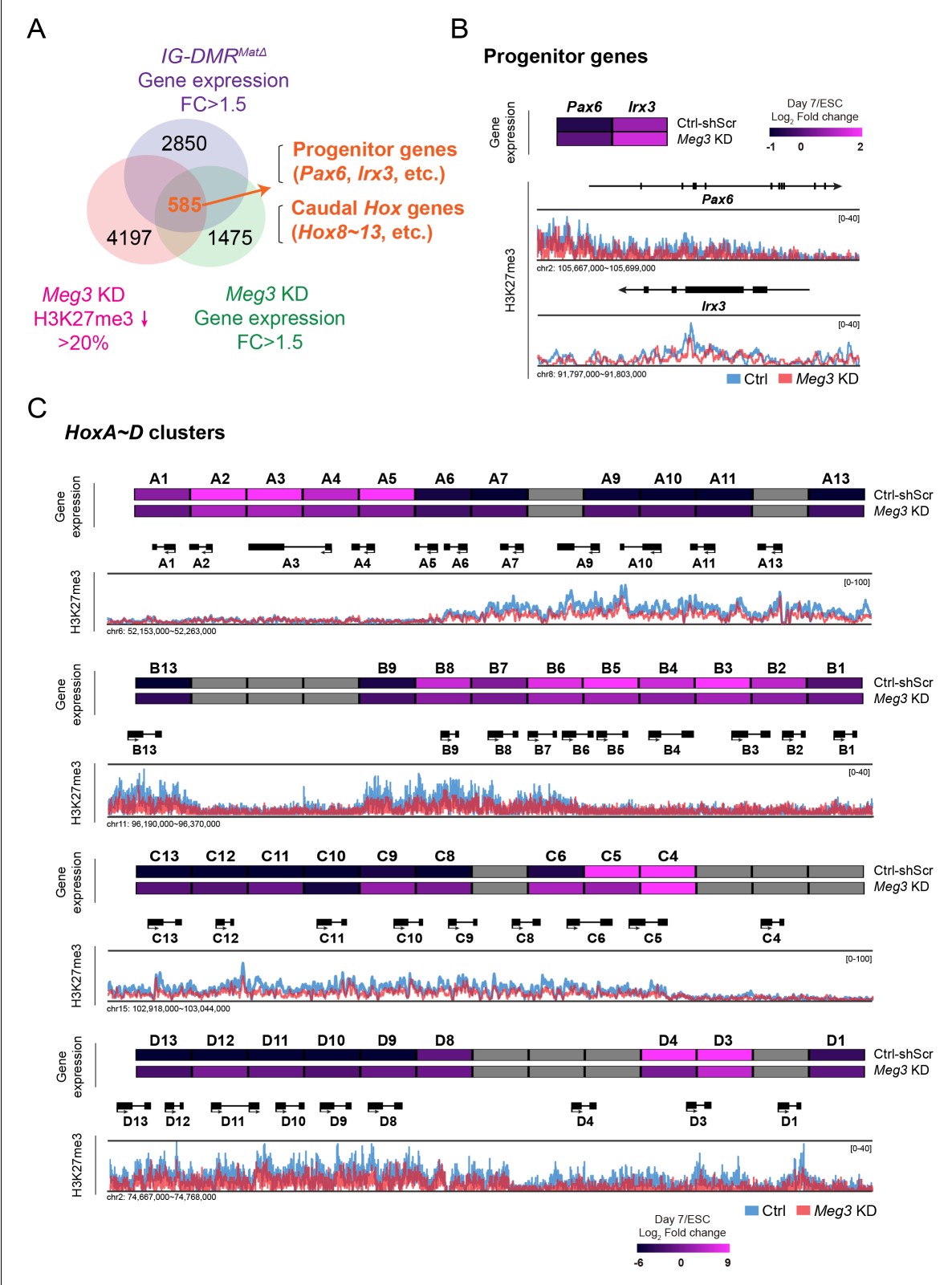

**Figure 4.** Loss of *Dlk1-Dio3* locus-derived lncRNAs in MNs leads to dysregulation of neural progenitor and caudal *Hox* genes. (**A**) Genome-wide profiling of H3K27me3 by ChIP-seq and gene expression by Agilent microarray in control, *IG-DMR^{matΔ}*, and *Meg3* KD ESC~MNs. Venn diagram shows the number of genes that are up-regulated in both *IG-DMR^{matΔ}* and *Meg3* KD MNs while also displaying the reduction of H3K27me3 epigenetic landscape. Loss of *Meg3* imprinted lncRNAs is related to the up-regulation of progenitor genes, as well as *Hox* genes. (**B and C**) Heatmaps illustrating

*Figure 4 continued on next page*

*Figure 4 continued*

the expression profiles of ESC~MNs in control scrambled and *Meg3* KD lines. The color indicates the $\log_2$ fold-change in signal intensity between ESCs and ESC~MNs. Genes in light grey are not represented in the microarrays. Loss of *Meg3* imprinted lncRNAs results in ectopic expression of progenitor genes in (B), and a majority of caudal *Hox* genes (*Hox8~13*) in (C), with concomitant down-regulation of H3K27me3 levels.

The online version of this article includes the following figure supplement(s) for figure 4:

**Figure supplement 1.** Characterization of *Meg3* KD ESCs.
**Figure supplement 2.** Ezh2 binding and the H3K27me3 landscape in ESC~MNs.

Subsequently, we observed that MN progenitor (*Pax6* and *Irx3*) and majority of caudal *Hox* genes (*Hox8~13*) were up-regulated with a concomitant down-regulation of the H3K27me3 epigenetic landscape (*Figure 4B and C*). We corroborated this finding by generating a third *Meg3* KD ESC line (I6) and confirming that all *Meg3* KD ESC~MNs exhibited imbalanced expression of 3' and 5' *Hox* genes across entire *Hox* clusters (*Figure 4—figure supplement 1E*). Thus, for both *Meg3* KD and *IG-DMR*$^{mat\Delta}$ ESC~MNs, dysregulation of progenitor and caudal *Hox* gene expression is apparent, likely due to loss of the robustness of the epigenetic landscape of postmitotic MNs.

If *Meg3* scaffolds PRC2/Jaird2 to maintain the silenced H3K27me3 epigenetic landscape in progenitor and caudal *Hox* genes of postmitotic MNs, we predicted that (1) the binding patterns of Ezh2/H3K27me3 would display concordant tendency in MNs; and (2) the bindings of Ezh2/Jarid2 to the gene loci of progenitor and caudal *Hox* genes in MNs would be compromised upon the loss of *Meg3*. Consistent with our prediction, we verified that (1) the epigenetic landscapes of H3K27me3 in the progenitor and caudal *Hox* genes uncovered here are concordant with Ezh2 enrichment revealed by a previous study that used the same ESC~MN differentiation approach to generate cervical Hox-a5$^{on}$ MNs (*Figure 4—figure supplement 2A,B*) (*Narendra et al., 2015*); (2) Upon *Meg3* KD, the bindings of Ezh2/Jaird2 to progenitor (i.e., *Pax6* and *Irx3*) and caudal *Hox* (i.e., *Hoxc8*) genes were concomitantly reduced (*Figure 4—figure supplement 2C–E*). Taken all together, these results suggest that *Meg3* bridges the PRC2/Jarid2 complex to perpetuate the rostral MN cell fate by silencing epigenetic state of MN progenitor and caudal *Hox* genes.

## *IG-DMR*$^{mat\Delta}$ embryos manifest dysregulation of progenitor genes in postmitotic MNs

To corroborate the observed phenotype of *IG-DMR*$^{mat\Delta}$ ESC~MNs, we further scrutinized the MN phenotype in *IG-DMR*$^{mat\Delta}$ embryos. Consistent with previous studies, *IG-DMR*$^{mat\Delta}$ embryos died soon after E16 (*Lin et al., 2007*), so we analyzed MN phenotypes from E10.5~E14.5 in this study. We first verified that the expression of *Meg3* is still lacking in the developing spinal cord of E14.5 *IG-DMR*$^{mat\Delta}$ embryos (*Figure 5—figure supplement 1A*). Ventral neuronal progenitor patterning was not affected in the *IG-DMR*$^{mat\Delta}$ embryos revealed by Olig2 and Nkx2.2 (*Figure 5—figure supplement 1B,C*). We then checked the dorsal progenitor proteins Pax6 and Irx3. Compared to the control littermates, we observed a significant increase in the percentage of Pax6$^{on}$ (45% penetrance, n = 5/11), and Irx3$^{on}$ cells (100% penetrance, n = 8/8) for postmitotic MNs (Is11/2$^{on}$) along the entire RC axis of the ventral spinal cord in the *IG-DMR*$^{mat\Delta}$ embryos (*Figure 5A and C*, only the cervical segment is shown; quantifications shown in *Figure 5B and D*). However, Hb9$^{on}$ and Isl1(2)$^{on}$ MNs were comparable between the control and *IG-DMR*$^{mat\Delta}$ embryos at E10.5 (*Figure 5E and F*). Although dorsal progenitor genes were aberrantly up-regulated in the postmitotic MNs, production of MNs remained relatively unaffected, suggesting that the generation of MNs is still intact with the co-expressed progenitor/postmitotic genes in the *IG-DMR*$^{mat\Delta}$ embryos (*Figure 5G*). This outcome is consistent with the postmitotic expression of *Meg3* and its function to maintain the silenced epigenetic state of progenitor genes.

## *IG-DMR*$^{mat\Delta}$ embryos display caudalized *Hox* genes in cervical MNs

Next, we checked if the expression of Hox proteins is affected in *IG-DMR*$^{mat\Delta}$ embryos. We first assessed how loss of *Dlk1-Dio3* locus-derived lncRNAs affected the specification of segmental MNs, marked by brachial (Hoxc6), thoracic (Hoxc9), and lumbar (Hoxd10) Hox levels. We observed comparable numbers of cells expressing respective Hox proteins at each segmental level between control and mutant embryos (*Figure 5—figure supplement 1D,E*). Columnar identities of axial (Lhx3$^{on}$) and

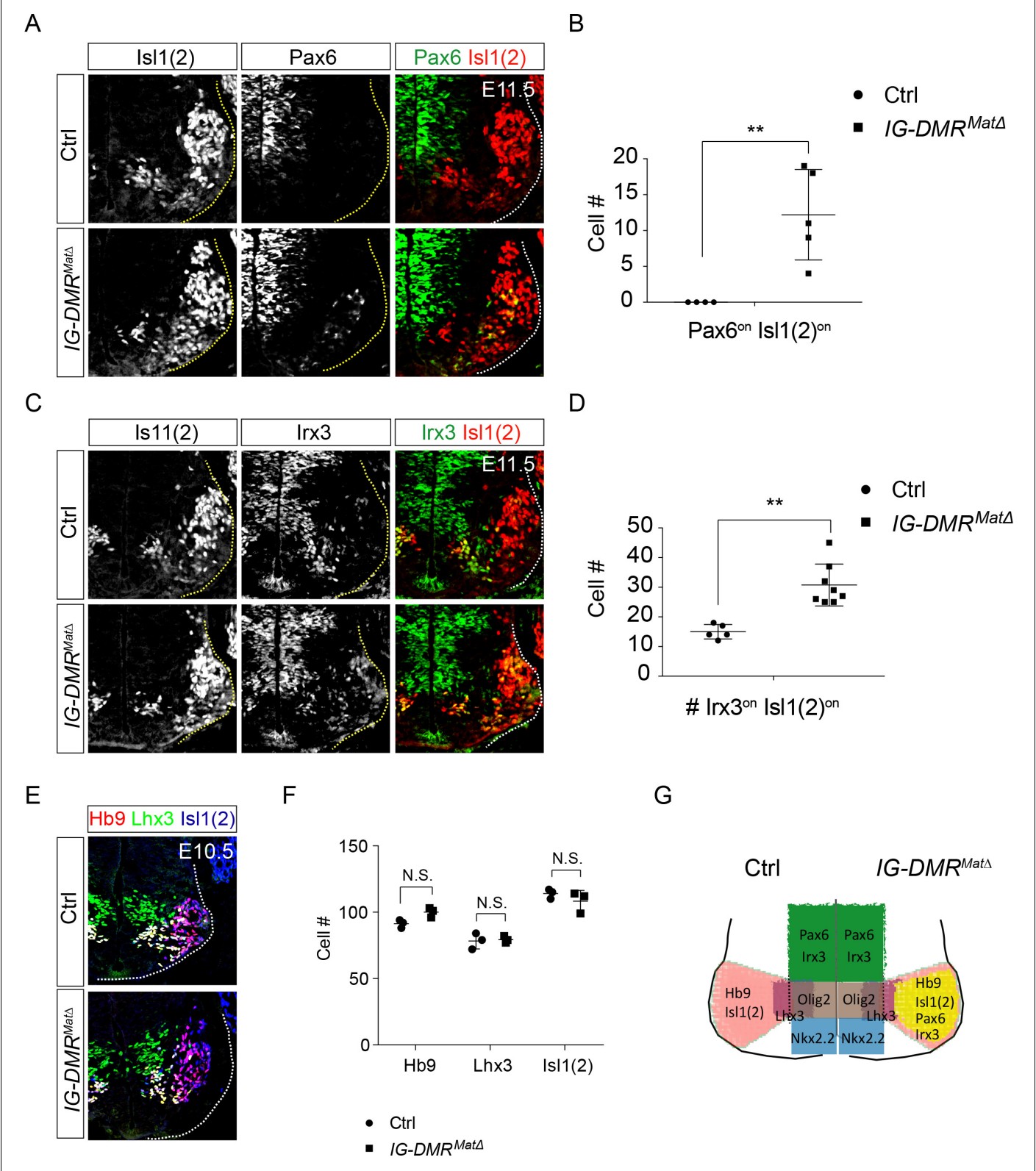

**Figure 5.** *IG-DMR^matΔ* mice ectopically turn on progenitor genes in postmitotic MNs. (A ~ D) *IG-DMR^matΔ* embryos display ectopic Pax6^on and Irx3^on cells in the postmitotic Is11(2)^on MNs (partial penetrance for Pax6, n = 5/11 at E11.5; whereas Irx3 displays 100% penetrance, n = 8/8. Error bars represent SD, ** p-value<0.01 by Student's *t*-test). (E) Generation of generic MNs (Hb9^on, Lhx3^on, and Isl1/2^on) is not affected in spinal cords of *IG-DMR^matΔ* mice at E10.5. (F) Quantification of postmitotic MNs (number of positive cells per 15 μm brachial ventral-half sections) in wild type control and

*Figure 5 continued on next page*

Figure 5 continued

IG-DMR$^{mat\Delta}$ embryos (error bars represent SD, n = 3 embryos at E10.5; N.S.: not significant by Student's *t*-test). (**G**) Summary of ectopic progenitor gene expression in postmitotic MNs of the IG-DMR$^{mat\Delta}$ embryos.

The online version of this article includes the following figure supplement(s) for figure 5:

**Figure supplement 1.** IG-DMR$^{mat\Delta}$ mutant phenotype analysis.

limb-innervating MNs (Foxp1$^{on}$) were also largely unaffected in the IG-DMR$^{mat\Delta}$ embryos (**Figure 5— figure supplement 1F,G**).

To further examine MN subtype diversification within the limb-innervating MNs, we checked the Hox proteins involved in pool specification (**Catela et al., 2016**; **Dasen et al., 2005**). Whereas reciprocal expression of Hoxa5 and Hoxc8 was maintained along the RC axis in the Hox6$^{on}$ LMC MNs of control embryos, Hoxc8 was expanded rostrally into Hoxa5$^{on}$ territory in IG-DMR$^{mat\Delta}$ embryos, along with a significant increase of the Hoxc8-mediated downstream motor pool genes, Pea3 and Scip (n = 5 embryos in **Figure 6A and C** for rostral brachial segments and **Figure 6B and D** for caudal brachial segments; quantification in **Figure 6E and F**). The reduction of Hoxa5 was not attributable to apoptosis, as cCasp3$^{on}$ cells were comparable in both control and IG-DMR$^{mat\Delta}$ mutant embryos (data not shown). Taken together, the switching of Hoxa5$^{on}$ to Hoxc8$^{on}$ in MNs of IG-DMR$^{mat\Delta}$ embryos is not transient, which leads to a concomitant change in motor pool fate (**Figure 6G**).

## Peripheral innervation defects in IG-DMR$^{mat\Delta}$ mutants

To further examine the impact of switching the MN pool subtype identity of LMC-MNs, we assessed the potential trajectory and target selectivity of motor axons in wild type control and IG-DMR$^{mat\Delta}$ embryos (**Liau et al., 2018**). We bred IG-DMR$^{mat\Delta}$ mutants to a transgenic line of *Hb9::GFP* mice in which all motor axons are labeled with GFP and then analyzed the overall pattern of limb innervation. First, the images of motor nerves from light sheet microscopy were converted into panoramic 3D images (upper panel in **Figure 7A**; **Videos 1** and **2**). The overall trajectory of each motor nerve was reconstructed by Imaris (lower panel in **Figure 7A**) and this conversion enabled semi-automatic calculation of the number of motor nerve terminals in each skeletal muscle, as well as comparison of the extent of motor axon arborization between skeletal muscles (see Materials and methods for details). Under higher magnification with better resolution, we observed the terminal arbors of suprascapular (Ss) nerves of scapulohumeralis posterior muscles and axillary (Ax) nerves were significantly eroded and reduced (**Figure 7B and C**), consistent with the caudalized switch from Hoxa5 to Hoxc8 expression within LMC neurons. Concomitantly, increased arborization complexity of distal muscle-innervating nerves, including posterior brachial cutaneous (PBC) nerves were manifested in the IG-DMR$^{mat\Delta}$ embryos (**Figure 7D**, quantification in 7E, n = 6, p<0.01, Mann–Whitney U test). Thus, the IG-DMR$^{mat\Delta}$ mutants displayed deficiencies in peripheral innervation of MNs, which might be a consequence of dysregulation of Hox proteins and/or other axon arborization genes in the absence of lncRNAs from the *Dlk1-Dio3* locus.

## Meg3 functions as a potent lncRNA in the Meg3-Rian-Mirg locus to regulate MN cell fate

As our current and previous results indicated that the expressions of most lncRNAs in the *Meg3-Rian-Mirg* locus are reduced upon *Meg3* KD and in IG-DMR$^{Mat\Delta}$ (**Figure 4—figure supplement 1A**) (**Lin et al., 2003**), it remains puzzling if these ncRNAs work independently or synergistically in this locus to regulate MNs. To further parse this question, we generated two single lncRNA *Rian$^{\Delta/\Delta}$* and *Mirg$^{\Delta/\Delta}$* ESCs respectively by using CRISPR-Cas9 mediated approaches (**Figure 8A**). The design of the targeted deletion regions of *Meg3* and *Rian* followed two previous studies (**Han et al., 2014**; **Labialle et al., 2014**), which led to a 23 kb deletion in the *Rian$^{\Delta/\Delta}$* ESC and a 20 kb deletion in the *Mirg$^{\Delta/\Delta}$* ESC (**Figure 8A**). We first verified that the paternal gene (i.e., *Dlk1*) is not affected in either *Rian$^{\Delta/\Delta}$* or *Mirg$^{\Delta/\Delta}$* ESCs. In the *Rian$^{\Delta/\Delta}$* ESC, expressions of *Meg3* and *Mirg* were relatively unaffected, whereas that of *Rian* was compromised. Conversely, only expression of *Mirg* was impaired significantly in the *Mirg$^{\Delta/\Delta}$* ESCs, but expressions of *Meg3* and *Rian* manifested minimal changes in that cell line (**Figure 8B**). Upon differentiation, *Meg3* KD ESC~MNs showed ectopic up-regulated expressions of progenitor and caudal *Hox* genes (**Figure 4** and **Figure 8C**). In contrast, expression

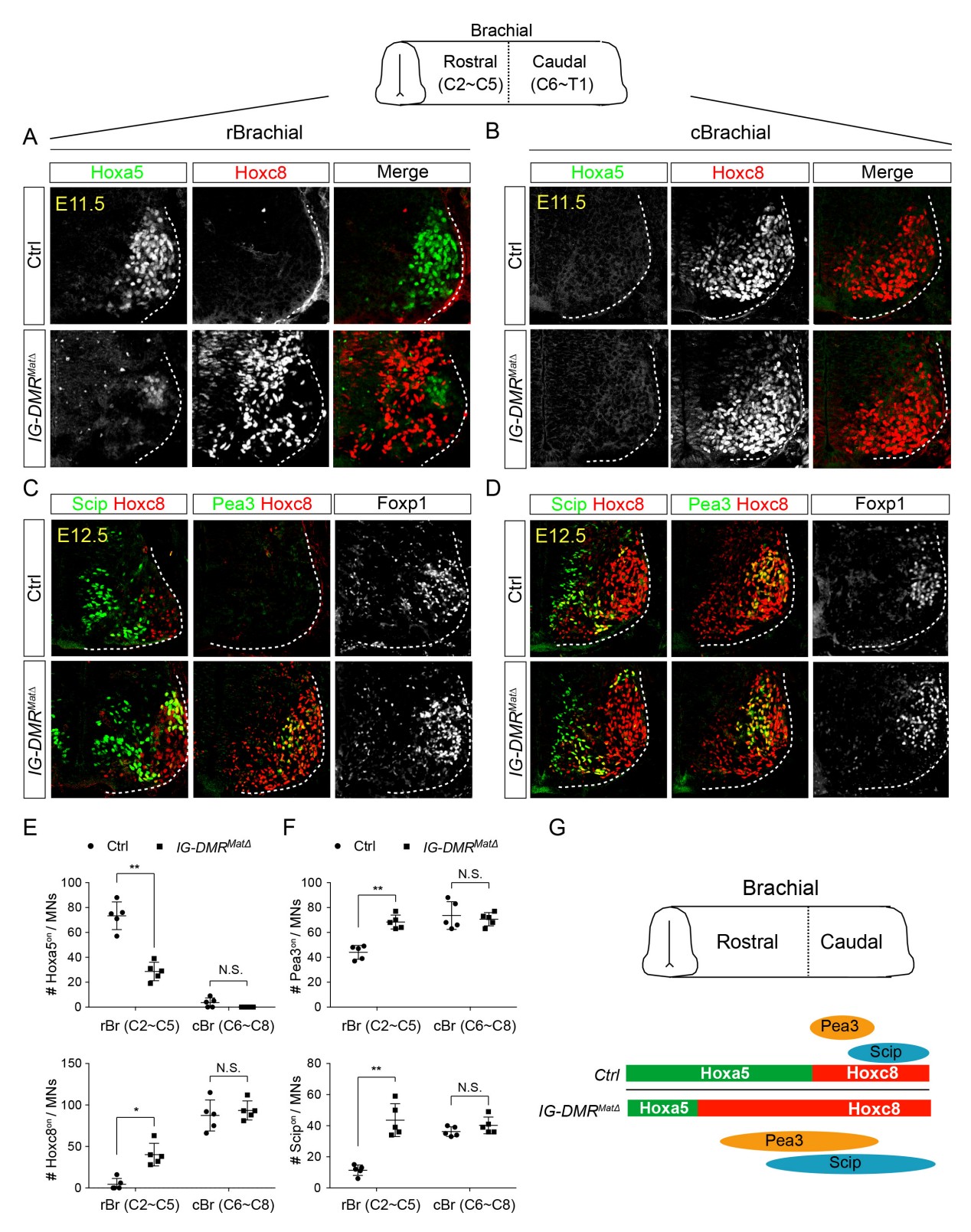

**Figure 6.** *IG-DMR^matΔ* mice manifest switched MN subtype identity. (A ~ D) Ectopic expansion of Hoxc8 and downstream Pea3^on and Scip^on MN pools in the rostral brachial segment, with a concomitant decrease of Hoxa5^on MNs for E11.5~E12.5 *IG-DMR^matΔ* embryonic spinal cord sections. Expression of Hoxc8 in the caudal brachial region remains unaffected (**B and D**). (**E and F**) Quantification of the numbers and distributions of Hoxa5^on, Hoxc8^on,

*Figure 6 continued on next page*

*Figure 6 continued*

Pea3[on], and Scip[on] MNs in the control and *IG-DMR^{matΔ}* mice from serial sections along the rostrocaudal axis (N.S.: not significant; error bars represent SD, n = 5; * p-value<0.05, ** p-value<0.01 by Student's *t*-test). (G) Summary of the motor neuron phenotype in the *IG-DMR^{matΔ}* embryos.

of *Hoxa5* remained unchanged in the *Rian^{Δ/Δ}* and *Mirg^{Δ/Δ}* ESC~MNs, and *Pax6* and *Hoxc8* expressions were also unaltered, with similar expression levels to controls (*Figure 8C*). These results indicate that *Meg3* might be the major regulatory lncRNA responsible for the observed MN phenotype displayed by the *IG-DMR^{matΔ}* embryos (*Figure 8D*).

## Discussion

Although mammalian genomes encode tens of thousands of lncRNAs, only less than a hundred have been shown to play critical roles in gene regulation in vitro. Consequently, the in vivo functions of the vast majority of lncRNAs remain to be vigorously tested. Strikingly, 40% of lncRNAs are expressed specifically in the central nervous system (CNS), which makes it one of the best systems for uncovering the physiological functions of lncRNAs (*Briggs et al., 2015*; *Ng et al., 2013*). In this study, we identified a series of novel and/or uncharacterized lncRNAs that exhibit precisely regulated temporal and spatial expression patterns during MN development. Here, we focused on characterizing lncRNAs located in the imprinted *Dlk1-Dio3* locus for three reasons: (1) this locus is conserved between human and mouse (*Lin et al., 2007*; *Lin et al., 2003*); (2) several studies have highlighted that *Meg3* might function as a tumor suppressor (*Zhang et al., 2010*; *Zhou et al., 2007*); and (3) a previous report indicates that the paternally-expressed coding gene *Dlk1* has an unexpected function in determining MN subtype diversification (*Müller et al., 2014*), which prompted us to examine whether the MN-enriched lncRNAs in the same locus also participate in MN cell fate determination.

### Functional perspectives of lncRNAs in MNs

Although lncRNAs derived from the *Dlk1-Dio3* locus are highly expressed in the CNS, their functions during neural development are largely unknown (*Wang et al., 2012*; *Zhang et al., 2003*; *Zhou et al., 2012*). Upon KD of *Meg3* (a *Dlk1-Dio3* locus-derived lncRNA), we uncovered that: 1) many adjacent progenitor genes were significantly up-regulated; and 2) the rostral *Hox* genes were significantly down-regulated, with a concomitant increased expression of caudal *Hox* genes in ESC~MNs. This phenotype was recapitulated in *IG-DMR^{matΔ}* embryos, in which Hoxc8 expression is expanded in otherwise Hoxa5[on] MNs. Several reports have identified that certain lncRNAs can shape the *Hox* epigenetic landscape by *cis* and *trans* modulation (*Dasen, 2013*; *Rinn et al., 2007*; *Wang et al., 2011*). In addition, we recently uncovered that a novel *trans* Hox-miRNA circuit can filter *Hox* transcription noise and prevents precocious protein expression to confer robust individual MN identity (*Li et al., 2017*). Here, we have now added the *Meg3* imprinted lncRNA to that list as a novel *trans*-acting lncRNA that maintains the *Hox* epigenetic landscape, most likely by recruiting Jarid2 to the PRC2 complex. Given that *Meg3* is also highly expressed in ESCs (*Kaneko et al., 2014a*; *Mo et al., 2015*), we plan to generate a targeted *Meg3* floxed allele mouse line in the future that will allow us to specifically knockout *Meg3* in MNs and recover the potential function of *Meg3* in cell-type specific contexts.

### A fail-safe mechanism to guard MN epigenetic landscape

Why do MNs deploy lncRNA-mediated strategy to maintain postmitotic cell fate by inhibiting progenitor genes and regulating Hox boundaries? The dynamic role of lncRNAs in modulating PRC2 function is well documented; ranging from recruitment, complex loading and activity control to gene targeting (*Davidovich and Cech, 2015*; *Kretz and Meister, 2014*). A recent study of *Drosophila Hox* genes revealed that epigenetic H3K27me3 chromatin modification functions as a legitimate carrier of epigenetic memory (*Coleman and Struhl, 2017*), providing compelling evidence for a physiologically significant role of chromatin modification in epigenetic inheritance. Nonetheless, the epigenetic memory carried by H3K27me3 in a postmitotic cell may still be overridden by H3K27me3-opposing demethylases (*Coleman and Struhl, 2017*). Our RNA-seq data revealed that two prominent H3K27me3 demethylases, *Kdm6a (Utx)* and *Kdm6b (Jmjd3)*, are reactivated in

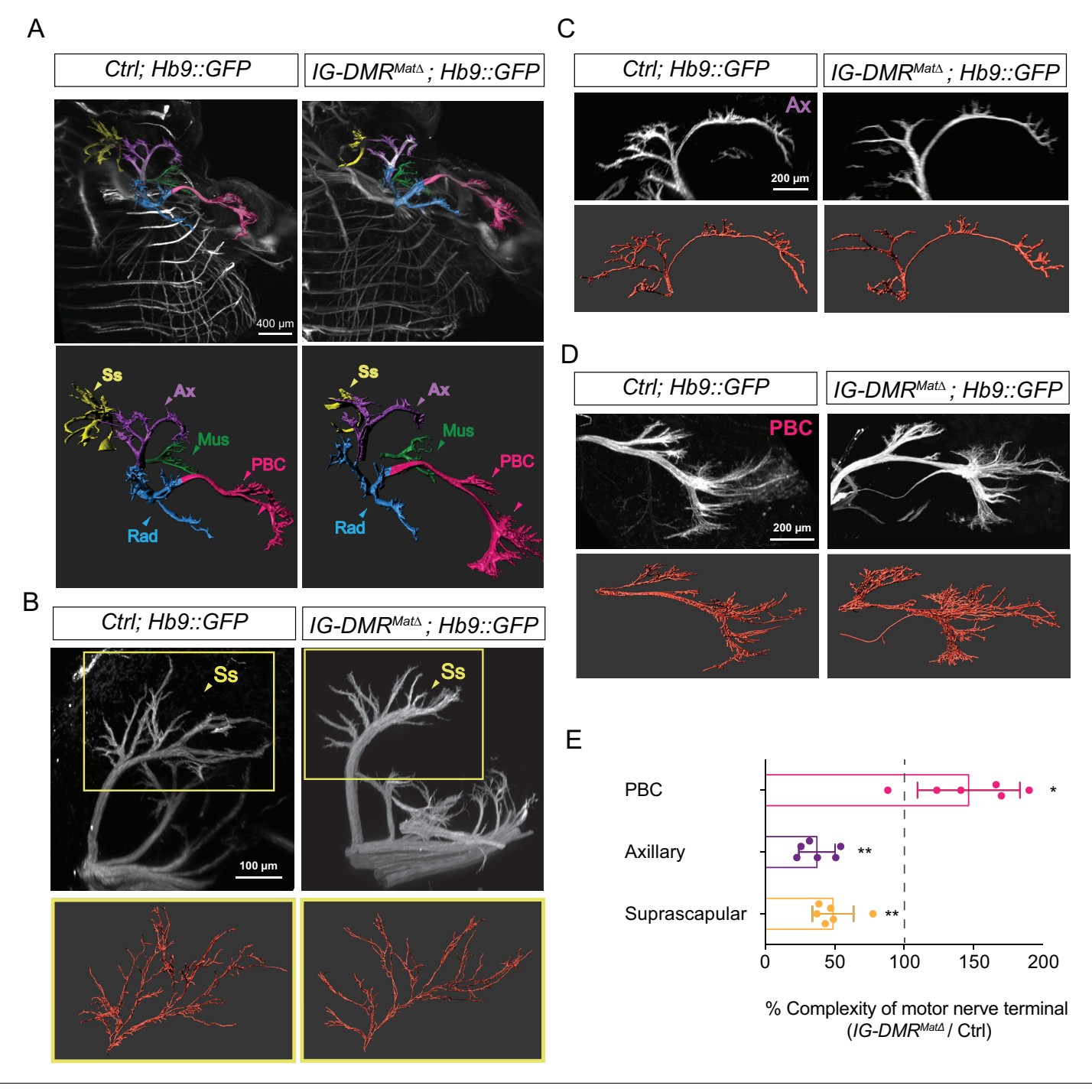

**Figure 7.** Motor axon innervation defects in the *IG-DMR^matΔ* embryos.  (**A**) Panoramic views from different angles of control and *IG-DMR^matΔ*; *Hb9::GFP* embryos at E13.5 using light sheet microscopy (Upper panel). Details of each viewing angle are illustrated in *Videos 1* and *2*. Reconstruction of motor nerve positions by Imaris (see Materials and methods for details) is illustrated in the lower panel. Suprascapular nerve (Ss, yellow); axillary nerve (Ax, purple); musculocutaneous nerve (Mus, green); radial nerve (Rad, blue); posterior brachial cutaneous nerve (PBC, pink). (B ~ D) Higher magnification of MN innervations in the forelimbs of E13.5 control and *IG-DMR^matΔ*; *Hb9::GFP* mice. Mutant mice display defects in Ss and Ax axonal branching, concomitant with more PBC axonal branching. Semi-automatic highlighting of the axonal branching and nerve trajectories is used and quantified by Imaris. (**E**) Quantification of the axonal branching and nerve trajectories for E13.5 control and *IG-DMR^matΔ*; *Hb9::GFP* mice by Imaris (see Materials and ethods for details) (n = 6, p<0.01, Mann–Whitney U test).

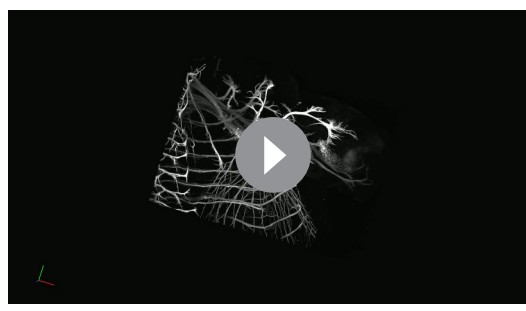

**Video 1.** Panoramic views from different angles of forelimb in control *Hb9::GFP* embryos at E13.5 using light sheet microscopy.
https://elifesciences.org/articles/38080#video1

postmitotic MNs (data not shown). Although the function of H3K27me3 demethylase reactivation in postmitotic MNs remains unknown, these findings raise the possibility that the epigenetic memory of the H3K27me3 landscape in postmitotic MNs might still need to be 'actively' maintained to counterbalance H3K27me3 demethylase activity. Enrichment of postmitotic MN lncRNAs might therefore bridge and scaffold the PRC2/Jarid2 interaction and activity to maintain MN epigenetic memory by repressing progenitor genes and also carve MN subtype identity by repressing caudal *Hox* genes (*Figure 8D*).

Although we observed a complete 100% penetrance of Hoxa5-c8 cell fate switch in the rostro-brachial segments and an increase co-expression of Irx3[on]Hb9[on] cells in the *IG-DMR[matΔ]* embryos (*Figures 5* and *6*), ectopic Pax6 was manifested at partial penetrance (45%, 5 in the 11 mutants) and several caudal Hox protein (including Hox9 and 10) were not shown to display significant change in vivo (*Figure 5* and *Figure 5—figure supplement 1*), consistent with the finding that removal of *Ezh2* from MN progenitors has no detectable impact on segmental Hox expression in the spinal cord (Hoxc6 and Hoxc9) (*Golden and Dasen, 2012*). This result also suggests that a compensatory PRC1-mediated function in vivo might make up for the loss of *Meg3*-mediated epigenetic maintenance in segmental MNs (*Golden and Dasen, 2012*; *Mazzoni et al., 2013b*). Our results are not entirely unexpected as many potent miRNA/lncRNA KO mice reflect either only partial phenotype penetrance (*Li et al., 2013*; *Li et al., 2016*; *Medeiros et al., 2011*), or more severe phenotypes upon genetic or environmental stresses (*Williams et al., 2009*). Moreover, many miRNAs/TFs function as repressors to silence progenitor/neighboring interneuron genes (*Chen et al., 2011*; *Shirasaki and Pfaff, 2002*), they may also constitute a coherent loop with lncRNAs to safeguard the terminal postmitotic cell fate with a fail-safe control.

## PRC2-lncRNA regulation in MNs

A previous study reported that hypomorphic *Suz12[-/-]* ESCs maintained with a low amount of H3K27me3 can differentiate into MNs, albeit with a significant increase in the expression of caudal *Hoxc6* and *Hoxa7* compared to wild-type cells (*Mazzoni et al., 2013b*). Interestingly, another study found that several PRC2 mutant ESC lines that maintain varying levels of H3K27me3 allowed for proper temporal activation of lineage genes during directed differentiation of ESCs to MNs, but only a subset of the genes that function to specify other lineages were not repressed in these cells

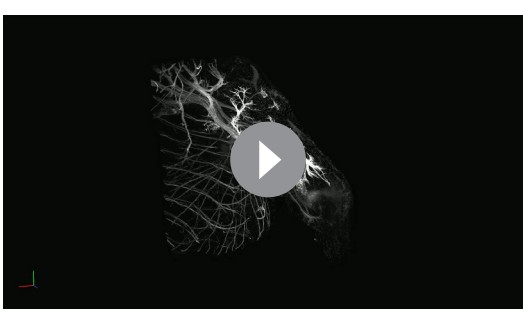

**Video 2.** Panoramic views from different angles of forelimb in *IG-DMR[matΔ]* ; *Hb9::GFP* embryos at E13.5 using light sheet microscopy.
https://elifesciences.org/articles/38080#video2

(*Thornton et al., 2014*). This outcome might not be surprising since other epigenetic marks, such as DNA methylation, might safeguard gene expression throughout differentiation (*Manzo et al., 2017*). In this study, up-regulated genes in spinal MNs upon loss of *Meg3-Rian-Mirg* exhibited 50% concordance (3953/7474) with the up-regulated genes in *Suz12[-/-]* spinal MNs (data not shown). Together, these results strongly endorse the critical function of PRC2/lncRNA in perpetuating the postmitotic cell fate of cervical Hoxa5[on] MNs.

Given that lncRNAs such as *Hotair* are proposed to scaffold the PRC2 complex and guide it to specific genome loci (*Rinn et al., 2007*; *Rinn and Chang, 2012*; *Tsai et al., 2010*), it is tantalizing to hypothesize that *Meg3* might

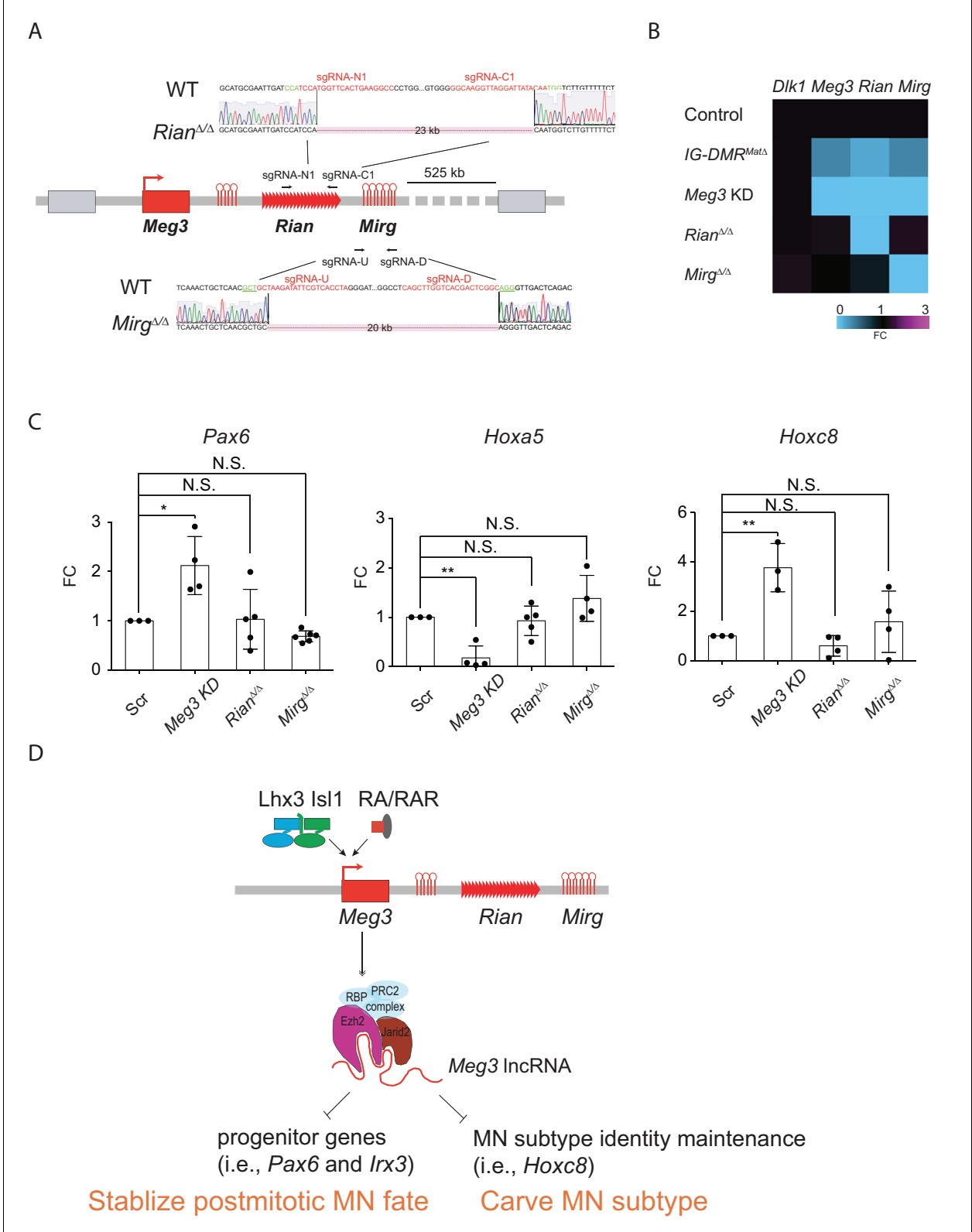

**Figure 8.** Dissection of individual roles of lncRNAs in the *Meg3-Rian-Mirg* locus in ESC~MNs. (**A**) Illustration of the sgRNAs target sites and sequences for the sgRNA:Cas9-mediated *Rian* and *Mirg* deletions respectively. Deleted sequences shown in *Rian*$^{\Delta/\Delta}$ and *Mirg*$^{\Delta/\Delta}$ ESC lines. The PAM sites are displayed in green; the sgRNA target sequences are reflected in red. (**B**) Heatmaps representing the abundances of the paternal gene (*Dlk1*) and maternal lncRNAs (*Meg3*, *Rian*, and *Mirg*) in control, *Meg3* KD, and KOs (*IG-DMR*$^{mat\Delta}$, *Rian*$^{\Delta/\Delta}$, and *Mirg*$^{\Delta/\Delta}$) ESCs respectively. (**C**) Rostral *Hox* gene

*Figure 8 continued on next page*

*Figure 8 continued*

*Hoxa5* is significantly down-regulated in the *Meg3* KD line, whereas the expression of *Hoxa5* is comparable between *Rian*$^{\Delta/\Delta}$ and *Mirg*$^{\Delta/\Delta}$ when compared to the controls. Conversely, the expressions of progenitor gene *Pax6*, as well as caudal *Hox* gene *Hoxc8*, are significantly up-regulated in the *Meg3* KD line, while their expressions are restored between *Rian*$^{\Delta/\Delta}$ and *Mirg*$^{\Delta/\Delta}$ when compared to the controls. (FC: fold-change; error bars represent SD, n = 3~5, * p-value<0.05, ** p-value<0.01 by Student's *t*-test). (D) Summary of the functions of lncRNAs from the imprinted *Dlk1-Dio3* locus in ESC~MNs. *Meg3* and other lncRNAs from the *Dlk1-Dio3* locus are directly activated by MN-TFs (i.e., Lhx3 and Isl1) and RAR, leading to enrichment of *Meg3* in the rostral segment of postmitotic MNs. One major function of *Meg3* and other lncRNAs from the *Dlk1-Dio3* locus is to stimulate Ezh2/Jarid2 interactions. Loss of these lncRNAs compromises the H3K27me3 epigenetic landscape and leads to aberrant expressions of progenitor and caudal *Hox* genes in postmitotic MNs. Our model illustrates that the lncRNAs of the imprinted *Dlk1-Dio3* locus (*Meg3* particularly) play a critical role in maintaining postmitotic MN cell fate by repressing progenitor genes, and that they shape MN subtype identity by regulating *Hox* genes.

manifest the dual functions of scaffolding the PRC2/Jarid2 complex and guiding it to specific loci in different cell contexts. This scenario could partially explain why only subsets of genes in an MN context are particularly sensitive to the loss of ncRNAs from the *Dlk1-Dio3* locus. Inspired by the salient Hox phenotype exhibited in the *IG-DMR*$^{mat\Delta}$ mutants, we envisage using *Meg3* as a paradigm to decipher the *Meg3*-protein-DNA interactome by ChIRP-seq/ChIRP-MS, thereby allowing us to decipher the detailed targeting mechanism of PRC2/Jarid2 involved in maintaining the epigenetic landscape during embryonic development (*Chu et al., 2015*). Interestingly, *Hotair* $^{-/-}$ mice also show partial penetrance of homeotic transformation and increased expression of *Meg3* (*Li et al., 2013*). This finding prompts us to test in the future the hypothesis that *trans*-acting lncRNAs might have an unexpected redundant role for PRC2 complex scaffolding and targeting despite having no primary sequence conservation. Generating compound lncRNA mutants that can scaffold the PRC2/Jarid2 complex will shed light on this topic.

## Regulatory mode of *Meg3* for the PRC2/Jarid2 complex

The role of lncRNAs in modulating PRC2 function is well documented, but very dynamic; from recruitment, complex loading and activity control to gene targeting (*Davidovich and Cech, 2015*; *Kretz and Meister, 2014*). Mammalian PRC2 binds thousands of RNAs in vivo, and it is a good system for studying the recruitment of chromatin modifying factors by RNA. Recent studies suggest that lncRNAs facilitate JARID2-PRC2 interactions on chromatin and propose a mechanism by which lncRNAs contribute to PRC2 recruitment (*da Rocha et al., 2014*; *Kaneko et al., 2014a*; *Kaneko et al., 2014b*). Other studies have provided an alternative working model, whereby the JARID2 and PRC2 sub-complex have different RNA-binding affinities and RNA binding to EZH2 inhibits its methyltransferase activity in a concentration- and binding affinity-dependent manner. Surprisingly, the binding of RNA attenuates the methyltransferase activity of EZH2, which allows JARID2 to relieve the repressive effect of RNA on PRC2 catalysis (*Cifuentes-Rojas et al., 2014*; *Davidovich et al., 2015*; *Davidovich et al., 2013*). Both these models differ but are not contradictory, as both models emphasize the role of RNA in the PRC2/JARID2 complex, albeit with different binding affinities and modes of action. Interestingly, a recent report further uncovered that *MEG3* binds to chromatin sites with GA/GT-rich sequences through RNA–DNA triplex formation (*Iyer et al., 2017*). This finding raises the possibility that the GA-rich motif alone is not sufficient in all cell types (*Chu et al., 2011*). Future systematic *Meg3* ChIRP-seq analyses, together with Ezh2/Jarid2 CLIP-seq, at all stages of ESC~MN differentiation might aid in identifying the context-dependent/independent lncRNA-mediated PRC2 targeting strategy.

## Combinatorial and individual roles of the *Meg3-Rian-Mirg* lncRNA cluster

There have been multiple efforts to dissect the functions of the maternally-expressed lncRNAs in the *Meg3-Rian-Mirg* locus over the past decade (*McMurray and Schmidt, 2012*; *Steshina et al., 2006*; *Takahashi et al., 2009*; *Zhou et al., 2010*), but definitive results remain elusive. The difficulty is mainly attributable to two major hurdles; namely that 1) there are many DMRs that control imprinting status in upstream, promoter, and exon regions of *Meg3*, and 2) *Meg3* might function *in cis* to regulate its imprinting status (*Matsubara et al., 2015*; *Ogata and Kagami, 2016*). Specifically, two *Meg3* knockout mouse lines have previously been generated either by deletion of the first five exons plus approximately 300 bp of the adjacent upstream promoter region of *Meg3* (~5.9 kb)

(*Zhou et al., 2010*) or by deletion of ~10 kb that includes the *Meg3*-DMR region plus the first five exons of *Meg3* (*Takahashi et al., 2009*). Both of these *Meg3* KO lines also manifested loss of maternal *Rian* and *Mirg* lncRNA expression. However, whereas the ~5.9 kb *Meg3* KO line (*Zhou et al., 2010*) exhibited perinatal lethality, the ~10 kb *Meg3* KO line (*Takahashi et al., 2009*) presented a much milder phenotype in that the mice were born alive and lived up to 4 weeks after birth. We also found that expression of most, if not all, lncRNAs in the *Meg3-Rian-Mirg* locus are reduced upon *Meg3* KD and in *IG-DMR^{MatΔ}* mutants. Therefore, it remains technically challenging to obtain a specific *Meg3* KO without abrogating the expression levels of downstream lncRNAs.

Although we have further shown here that *Meg3* acts as a scaffold for the PRC2/Jarid2 complex, it is still possible that *Rian* and *Mirg* could independently and/or synergistically function with *Meg3* to contribute to the Hox-mediated MN subtype switching we observed in the *IG-DMR^{MatΔ}* mutants. To investigate this possibility, we instead generated two independent targeted deletion ESC lines of *Meg3* downstream lncRNAs, *Rian^{Δ/Δ}* and *Mirg^{Δ/Δ}*. Consistent with previous results (*Han et al., 2014*; *Labialle et al., 2014*), deletions of these downstream lncRNAs show less profound phenotype than upstream *Meg3*. Our results reveal that the progenitor gene *Pax6* and caudal *Hox* genes in these two KO ESC~MNs are relatively unaffected when compared to *Meg3* KD. Interestingly, *Mirg* KO embryos seem to have a less severe phenotype, and no observed homeotic transformation has been reported (*Labialle et al., 2014*). Furthermore, our results indicate that *Mirg* does not bind to Ezh2/Suz12/Jarid2 (*Figure 3A*). Thus, it is likely that *Meg3* might be the major contributor to MN subtype specification via epigenetic regulation. In this study, we did not delete lncRNA *Rtl1as* in the *Dlk1-Dio3* locus, as the deletion of *Rtl1as* simultaneously compromises paternal *Rtl1* expression (*da Rocha et al., 2008*). Therefore, we still can not completely rule out the possible synergistic ncRNA effects accounting for MN phenotype we observed in the *IG-DMR^{MatΔ}* embryos. Since lncRNAs are emerging as important modulators of gene regulatory networks and as epigenetic regulators of gene expression, we are endeavoring to systematically generate individual lncRNA KO embryos in the *Meg3-Rain-Mirg* locus by a CRISPR-Cas9-mediated approach. We anticipate that a detailed map of individual and synergetic lncRNA/miRNA functions during neural development attributable to this imprinted locus will be uncovered in the near future.

## Versatile functions of *Meg3* in development and disease

Consistent with the imprinting status of the *DLK1-DIO3* locus in humans, epimutations (hypermethylations) and microdeletions affecting *IG-DMR* and/or *MEG3-DMR* of maternal origin result in a unique human phenotype manifested as a small bell-shaped thorax, coat-hanger-like appearance of the ribs, abdominal wall defects, placentomegaly and polyhydramnios. One hallmark of patients with this disease, termed 'Kagami-Ogata syndrome' (KOS) (*Kagami et al., 2015*; *Ogata and Kagami, 2016*), is that nearly all of them display delayed gross motor development. It is currently unknown why epimutations and microdeletions of maternal *IG-DMR* give rise to this phenotype.

In our *IG-DMR^{matΔ}* embryos, we previously observed extra ossification at the sites where the 6th to 8th ribs attach to the sternum, similar to the malformed thorax in KOS patients (*Lin et al., 2007*). Interestingly, this phenotype was also observed in *Hox5* mutant mouse embryos (*McIntyre et al., 2007*). Here, we uncovered that two isoforms of the *Meg3* imprinted lncRNA are enriched in embryonic MNs and confer the fidelity of the epigenetic landscape for the Hoxa5-Hoxc8 boundary of MN subtypes. Loss of *Meg3* in vitro and in vivo abrogates the Hoxa5^{on} MNs in the brachial region, with a concomitant increase of ectopic Hoxc8^{on} subtypes. This switch leads to erosion of Hoxa5^{on} motor axon arborization in the proximal muscles. As *Meg3* expression is also highly enriched in somites, we suggest that impairment of the Hox boundary mediated by *Meg3* in the spinal cord and ribs might account for the bell-shaped thorax and motor deficit in KOS patients, potentially identifying a new therapeutic target for KOS patients.

In addition to the roles of the *Meg3* imprinted lncRNA uncovered by our study, other reports have also emphasized *Meg3* as being important for proper growth and development and to be a putative tumor suppressor that activates p53 and inhibits cell proliferation (*Takahashi et al., 2009*; *Zhang et al., 2010*; *Zhou et al., 2007*). Moreover, aberrant repression of *Meg3* and other maternally-expressed lncRNAs from the *DLK1-DIO3* imprinting cluster is present in several induced pluripotent stem cell (iPSC) lines. This scenario might lead to failure of these iPSCs to form viable mice (*Stadtfeld et al., 2010*) or to efficiently differentiate into neural lineage cells (*Mo et al., 2015*), raising the possibility that *Meg3* might be involved in a broad spectrum of developmental processes

and disease contexts. Thus, our exploration of *Meg3* may also suggest new avenues for treating other diseases, such as cancers, as well as in elucidating the reprogramming mechanism of iPSCs.

# Materials and methods

## Key resources table

| Reagent type (species) or resource | Designation | Source or reference | Identifiers | Additional information |
|---|---|---|---|---|
| Antibody | Rabbit polyclonal anti-Lhx3 | Abcam | Cat# ab14555, RRID:AB_301332 | ICC (1:2000) |
| Antibody | Rabbit polyclonal anti-Foxp1 | Abcam | Cat# ab16645, RRID:AB_732428 | ICC (1:20000) |
| Antibody | Mouse monoclonal anti-Isl1(2) | DSHB | Cat# 39.4D5, RRID:AB_2314683 | ICC (1:1000) |
| Antibody | Rabbit polyclonal anti-Hoxa5 | Sigma-Aldrich | Cat# HPA029319, RRID:AB_10601430 | ICC (1:5000) |
| Antibody | Rabbit polyclonal anti-Hoxa5 | Jeremy Dasen (NYU) | | ICC (1:16000) |
| Antibody | Guinea pig polyclonal pig anti-Hoxa5 | Jun-An Chen (Academia Sinica) | AB_2744661 | ICC (1:20000) |
| Antibody | Mouse monoclonal anti-Hoxc8 | DSHB | Cat# PCRP-HOXC8-1D3, RRID:AB_2618723 | ICC (1:1000) |
| Antibody | Rabbit polyclonal anti-Hoxc8 | Sigma-Aldrich | Cat# HPA028911, RRID:AB_10602236 | ICC (1:5000) |
| Antibody | Rabbit polyclonal anti-Pea3 | Thomas Jessell (Columbia University) | Cat# C115, RRID:AB_2631446 | ICC (1:2000) |
| Antibody | Guinea pig polyclonal anti-Scip | Thomas Jessell (Columbia University) | Cat# CU822, RRID:AB_2631303 | ICC (1:2000) |
| Antibody | Sheep polyclonal anti-GFP | AbD Serotec | Cat# 4745–1051, RRID:AB_619712 | ICC (1:1000) |
| Antibody | Rabbit polyclonal anti-Irx3 | Thomas Jessell (Columbia University) | | ICC (1:16000) |
| Antibody | Rabbit anti-Pax6 | Covance | Cat# PRB-278P, RRID:AB_291612 | ICC (1:300) |
| Antibody | Guinea pig polyclonal anti-Hb9 | Hynek Wichterle (Columbia University) | | ICC (1:1000) |
| Antibody | Mouse monoclonal anti-Hb9 | DSHB | Cat# 81.5C10, RRID:AB_2145209 | ICC (1:200) |
| Antibody | Mouse monoclonal anti-Nkx2.2 | DSHB | Cat# 74.5A5, RRID:AB_53179 | ICC (1:100) |
| Antibody | Rabbit polyclonal anti-Olig2 | Millipore | Cat# AB9610, RRID:AB_570666 | ICC (1: 20000) |
| Antibody | Goat polyclonal anti-Hoxc6 | Santa Cruz | Cat# sc-46135, RRID:AB_2119751 | ICC (1:1000) |
| Antibody | Rabbit polyclonal anti-Hoxc9 | Thomas Jessell (Columbia University) | Cat# Rabbit one anti-HoxC9, RRID:AB_2631312 | ICC (1:1000) |
| Antibody | Goat polyclonal anti-Hoxd10 | Santa Cruz | Cat# sc-33005, RRID:AB_648462 | ICC (1:1000) |
| Antibody | Mouse monoclonal anti-Ezh2 | Millipore | Cat# 17–662, RRID:AB_1977568 | WB (1:2000); RIP/ChIP/IP (2 μg) |
| Antibody | Rabbit polyclonal anti-Jarid2 | Novus | Cat# NB10 0-2214, RRID:AB_10000529 | WB (1:1000); RIP/ChIP (5 μg) |

*Continued on next page*

*Continued*

| Reagent type (species) or resource | Designation | Source or reference | Identifiers | Additional information |
|---|---|---|---|---|
| Antibody | Rabbit polyclonal anti-Suz12 | Abcam | Cat# ab12073, RRID:AB_442939 | WB (1:3000); RIP (2.5 µg) |
| Antibody | Mouse monoclonal anti-Histone H3 | Abcam | Cat# ab24834, RRID:AB_470335 | WB (1:1000) |
| Antibody | Mouse monoclonal anti-H3K27me3 | Abcam | Cat# ab6002, RRID:AB_305237 | ChIP (2 µg) |
| cell line | Mouse: *Hb9::GFP* ESC | PMID:12176325 | | Dr. Hynek Wichterle (Columbia University) |
| Cell line | Mouse: *Meg3* KD H6 ESC | this paper | | |
| Cell line | Mouse: *Meg3* KD K4 ESC | this paper | | |
| Cell line | Mouse: *Meg3* KD I6 ESC | this paper | | |
| Cell line | Mouse: *iMeg3*^v1 ESC | this paper | | |
| Cell line | Mouse: *iMeg3*^v5 ESC | this paper | | |
| Cell line | Mouse: *IG-DMR*^matΔ ESC | this paper | | |
| Cell line | Mouse: *Rian*^Δ/Δ ESC | this paper | | |
| Cell line | Mouse: *Mirg*^Δ/Δ ESC | this paper | | |
| Genetic reagent (*M. musculus*) | *Hb9::GFP* | PMID:12176325 | | Dr. Hynek Wichterle (Columbia University) |
| Genetic reagent (*M. musculus*) | *IG-DMR*^matΔ | PMID:12937418 | | Dr. Ann Fergusson Smith (University of Cambridge) |
| Commercial assay or kit | Biotin RNA Labeling Mix | Roche | 11685597910 | |
| Commercial assay or kit | Oligo Clean and Concentrator | Zymo Research | D4060 | |
| Commercial assay or kit | RNA-Binding Protein Immunoprecipitation Kit | Millipore | 17–700 | |
| Commercial assay or kit | TruSeq ChIP Library Preparation Kit | Illumina | IP-202–1012 | |
| Software, algorithm | Imaris 8.4.0 | | RRID:SCR_007370 | |

***Meg3* in situ hybridization probe primer**

| Gene | Forward primer | Reverse primer |
|---|---|---|
| *Meg3* | GAGTAATACGACTCACTATAGGGAACGTGTTGTGCGTGAAGTC | AACGTGTTGTGCGTGAAGTC |

**List of primers for quantitative RT-PCR analyses**

| Gene name | Forward primer | Reverse primer |
|---|---|---|
| *Meg3*^v1 | GCTTCTCGAGGCCTGTCTAC | GAACCTGAGCACAACAGCAA |
| *Meg3*^v5 | GAGGGACAAGCGACAAAGAG | CAGATGAACACGAGCACAGA |
| *Meg3*^v1(exon1–5) | GCTGCTTTCCTTCCTCACCT | TTTTCTCCTCAGCCCTTTGA |
| *Meg3*^v1(exon1–2) | GCTTCTCGAGGCCTGTCTAC | GAACCTGAGCACAACAGCAA |
| *Meg3*^v5(exon2–3) | TGCACCTCTACCTCCTGAGC | CAAGGTTTGAACCCCAGAGA |
| *Meg3*^v5–v1-1 | GCCCAAGTCTGGTAGCATGT | TACCTCAGCCATAGCCTGGT |
| *Meg3*^v5–v1-2 | GGGTGAATTGGCATTGATTT | GGGGTAGACAACCTGGCTTT |
| *Rian* | CTTCCAGGGTGAATTTTCCTAA | TATGGCCAAGCAATTCTGC |
| *Rian*^sgΔ | TGGATATCCTGCAAGTCGGC | GAACAGAGCTGACCGTGACA |

*Continued on next page*

*Continued*

**List of primers for quantitative RT-PCR analyses**

| Gene name | Forward primer | Reverse primer |
|---|---|---|
| Mirg | ACGACAACCGACAACAAAGA | GAAAGCCAAGAGCAGAAACC |
| Dlk1 | CGGGAAATTCTGCGAAATAG | TGTGCAGGAGCATTCGTACT |
| Dio3 | GAGTCCTGCTGCTTTTGTGTT | CCCTCTTCCACCCCTTTTT |
| Hb9 | GTTGGAGCTGGAACACCAGT | GCTCTTTGGCCTTTTTGCT |
| Irx3 | GTCCAAGCGGGGAATTTG | AGCCCAAGATCTGGTCACTG |
| Pax6 | GGACTGAGCTGACCCAAGAG | CAAGAGGGGAGGGGAAGTAG |
| Hoxa1 | ACCAAGAAGCCTGTCGTTCC | TAGCCGTACTCTCCAACTTTCC |
| Hoxa2 | CCTGGATGAAGGAGAAGAAGG | GTTGGTGTACGCGGTTCTCA |
| Hoxa3 | TCAAGGCAGAACACTAAGCAGA | ATAGGTAGCGGTTGAAGTGGAA |
| Hoxa4 | TGTACCCCTGGATGAAGAAGAT | AAGACTTGCTGCCGGGTATAG |
| Hoxa5 | TGTACGTGGAAGTGTTCCTGTC | GTCACAGTTTTCGTCACAGAGC |
| Hoxa6 | ACCGACCGGAAGTACACAAG | AGGTAGCGGTTGAAGTGGAAT |
| Hoxa7 | GAAGCCAGTTTCCGCATCTAC | CTTCTCCAGTTCCAGCGTCT |
| Hoxa9 | TCCCTGACTGACTATGCTTGTG | ATCGCTTCTTCCGAGTGGAG |
| Hoxa10 | GAAGAAACGCTGCCCTTACAC | TTTCACTTGTCTGTCCGTGAG |
| Hoxa13 | GCTGCCCTACGGCTACTTC | GCGGTGTCCATGTACTTGTC |
| Hoxb1 | TTCGACTGGATGAAGGTCAA | GGTGAAGTTTGTGCGGAGAC |
| Hoxb2 | ACCAAGAAACCCAGCCAATC | AGCAGTTGCGTGTTGGTGTAG |
| Hoxb3 | CAACTCCACCCTCACCAAAC | ACCACAACCTTCTGCTGTGC |
| Hoxb4 | CTGGATGCGCAAAGTTCAC | GACCTGCTGGCGAGTGTAG |
| Hoxb5 | AGGGGCAGACTCCACAGATA | CTGGTAGCGAGTATAGGCGG |
| Hoxb6 | AGCAGAAGTGCTCCACGC | TAGCGTGTGTAGGTCTGGCG |
| Hoxb7 | CGAGAGTAACTTCCGGATCTACC | TTTCTCCAGCTCCAGGGTCT |
| Hoxb8 | ACACAGCTCTTTCCCTGGATG | GGTCTGGTAGCGACTGTAGGTC |
| Hoxb9 | AGGAAGCGAGGACAAAGAGAG | TGGTATTTGGTGTAGGGACAGC |
| Hoxb13 | ATTCTGGAAAGCAGCGTTTG | CTTGCTATAGGGAATGCGTTTT |
| Hoxc4 | AGCAAGCAACCCATAGTCTACC | GCGGTTGTAATGAAACTCTTTCTC |
| Hoxc5 | CACAGATTTACCCGTGGATGAC | CTTTCTCGAGTTCCAGGGTCT |
| Hoxc6 | TAGTTCTGAGCAGGGCAGGA | CGAGTTAGGTAGCGGTTGAAGT |
| Hoxc8 | GTAAATCCTCCGCCAACACTAA | CGCTTTCTGGTCAAATAAGGAT |
| Hoxc9 | GCAAGCACAAAGAGGAGAAGG | CGTCTGGTACTTGGTGTAGGG |
| Hoxc10 | CGGATAACGAAGCTAAAGAGGA | GCGTCTGGTGTTTAGTATAGGG |
| Hoxc12 | GACTCCAGTTCGTCCCTACTCA | TGAACTCGTTGACCAGAAACTC |
| Hoxc13 | GGAAGTCTCCCTTCCCAGAC | GCGTACTCCTTCTCTAGCTCCTT |
| Hoxd1 | ACAGCACTTTCGAGTGGATGA | AGGGCTTGTGGCTCCATATT |
| Hoxd3 | CTACCCTTGGATGAAGAAGGTG | TCAGACAGACACAGGGTGTGA |
| Hoxd4 | CTACCCTTGGATGAAGAAGGTG | TTCTAGGACTTGCTGTCTGGTG |
| Hoxd8 | GCTCGTCTCCTTCTCAAATGTT | GCGACTGTAGGTTTGTCTTCCT |
| Hoxd9 | CAGCAACTTGACCCAAACAAC | TGGTATTTGGTGTAGGGACAGC |
| Hoxd10 | CTGAGGTTTCCGTGTCCAGT | CAATTGCTGGTTGGAGTATCAG |
| Hoxd11 | ACACCAAGTACCAGATCCGC | AGTGAGGTTGAGCATCCGAG |
| Hoxd12 | CTTCACTGCCCGACGGTA | TGCTTTGTGTAGGGTTTCCTCT |
| Hoxd13 | GGAACAGCCAGGTGTACTGTG | GTAAGGCACCCTTTTCTTCCTT |

*Continued on next page*

*Continued*

**List of primers for quantitative RT-PCR analyses**

| Gene name | Forward primer | Reverse primer |
| --- | --- | --- |
| Neurog2 | GACATTCCCGGACACACAC | AGCTCCTCGTCCTCCTCCT |
| Isl1 | AGCTGGAGACCCTCTCAGTC | TGCTTCTCGTTGAGCACAGT |
| Gapdh | AGGCCGGTGCTGAGTATGTC | GCCTGCTTCACCACCTTCT |
| Rnu1 | GGGAGATACCATGATCACGAAGGT | CCACAAATTATGCAGTCGAGTTTCCC |
| Malat1 | CATGGCGGAATTGCTGGTA | CGTGCCAACAGCATAGCAGTA |
| Kcnq1ot1 | CCTTCCTTGTGCTTTGACCC | GATCGCCTAAGACCATCGGA |

**sgRNA sequences used in this work**

| Target | Sequence (5' ~ 3') |
| --- | --- |
| sg-Rian N1 | TCCATGGTTCACTGAAGGCC |
| sg-Rian C1 | GGCAAGGTTAGGATTATACAA |
| sg-Mirg U | GCTAAGATATTCGTCACCTA |
| sg-Mirg D | CAGCTTGGTCACGACTCGGC |

**List of primers for CRISPR deletion genotyping/sequencing**

| Gene name | Forward primer | Reverse primer |
| --- | --- | --- |
| Rian WT | AACCATGGCATCTGTGTGAA | CAAAAATCAACCGCCCTCTA |
| Rian KO | GATGTGACTGCTTTGAGGCA | GTGCTCCAGAAGCCGAAAAG |
| Mirg WT | CCACTCTCCTCAGCATCCAT | GAGCAGTTTGAGAGGCCCTA |
| Mirg KO | CCACTCTCCTCAGCATCCAT | GCTCTGGGGAGAACAGTGAG |

**Published ChIP-Seq data summary (related to *Figure 2D, G and H, Figure 4—figure supplement 1A,B*)**

| ChIP samples | GEO accession | Publications |
| --- | --- | --- |
| Induced V5-tagged Lhx3 (iLhx3-V5) in iNIL3-induced motor neurons (Day 4) | GSM782847 | (*Mazzoni et al., 2013a*) |
| Isl1/2 in iNIL3-induced motor neurons (Day 4) | GSM782848 | (*Mazzoni et al., 2013a*) |
| H3K4me3 | GSM1468401 | (*Narendra et al., 2015*) |
| H3K27ac_day6 | GSM2098385 | (*Rhee et al., 2016*) |
| ATAC_seq_day6 | GSM2098391 | (*Rhee et al., 2016*) |
| RAR_Day2 + 8hrsRA | GSM482750 | (*Mahony et al., 2011*) |
| Pol2-S5P_Day2 + 8 hr | GSM981593 | (*Mahony et al., 2011*) |
| H3K27me3.MN.WT | GSM1468398 | (*Narendra et al., 2015*) |
| EZH2.MN.WT | GSM1468404 | (*Narendra et al., 2015*) |

## Mouse ESC culture and MN differentiation

ESCs were cultured and differentiated into spinal MNs as previously described (*Wichterle et al., 2002*; *Wichterle et al., 2009*). Cells were trypsinized and collected for FACS at day seven to purify GFP$^{on}$ neurons for qPCR analysis and strand-specific RNA-seq when required. All cell lines used in this study are subject to regular mycoplasma test.

## Mouse crosses and in vivo studies

The *IG-DMR$^{matΔ}$* mouse strain is described in *Lin et al. (2003)*. Female mice carrying the deletion were mated with wild type C57BL6/J male mice to generate embryos with the maternally-inherited

deletion. Mice were mated at the age of 8~12 weeks and the embryo stage was estimated as E0.5 when a copulation plug was observed. Embryos were analyzed between E9.5~E13.5. All of the live animals were kept in an SPF animal facility, approved and overseen by IACUC Academia Sinica.

## Knockdown of *Meg3* in mouse ES cells by shRNA

The *Meg3* HuSH-29 shRNA plasmids (Origene, cat. No. TG501330) and non-effective scrambled sequence (TR20003) were used to create stable knockdown lines of *Meg3* within the *Hb9::GFP* ESCs. We used two different shRNA sequences to knockdown *Meg3*. Additionally, stable infected ESCs were selected by puromycin. Single ESC clones with good morphology and only presenting knockdown efficiencies >90% were picked for further expansion and characterization.

## Expression analysis

ESCs or embryoid bodies were harvested for total RNA isolation by Trizol (Thermo Scientific). For qPCR analysis, total RNA from each sample was reverse transcribed with Superscript III (Thermo Scientific). One-tenth of the reverse transcription reaction was used for subsequent qPCR reactions, which were performed in triplicate with three independent experimental samples on a LightCycler480 Real Time PCR instrument (Roche) using SYBR Green PCR mix (Roche) for each gene of interest. *Gapdh* was used as a control for normalization. For GeneChip expression analysis, RNA was purified and amplified using the Qiagen RNAeasy kit and a one-color Low Input Quick Amp Labeling Kit (Agilent Genomics) and hybridized to a SurePrint G3 Mouse GE 8×60K Microarray. Differentially-expressed genes were defined by ranking all probes according to *Moderated t-test* and a fold-change threshold ≥2 (p<0.001).

## Chromatin immunoprecipitation (ChIP)

We followed a previously published protocol to perform ChIP-seq for ESC~MNs (*Mazzoni et al., 2011*; *Narendra et al., 2015*). Four million cells were freshly dissociated from day 7 ESC~MNs by trypsin and fixed in 10 mM HEPES pH 7.6, 1% formaldehyde, 15 mM NaCl, 0.15 mM EDTA and 0.075 mM EGTA for 15 min at room temperature. After fixation, cells were quenched with 1.25 M glycine. After an ice-cold PBS wash and low-speed centrifugation, nuclear extracts were suspended with ice-cold shearing buffer (SDS included) containing protease inhibitor and sheared using a Covaris M220 system to an average chromatin size of 200 bp. Chromatin was diluted with 2X IP buffer (2% NP-40, 200 mM NaCl in 10 mM Tris-HCl pH 8, 1 mM EDTA). Anti-H3K27me3 antibody was added to each ChIP (antibodies are listed in the resource table). Each ChIP reaction was performed in a rotator at 4°C overnight, followed by washing in wash buffer (25 mM HEPES pH 7.6, 1 mM EDTA, 0.1% N-Lauryl sarcosine, 1% NP-40, and 0.5 M LiCl) at room temperature. Cross-linking was reversed at 65°C for 16 hr with 5 M NaCl. Proteinase K was added to digest for another 2 hr at 56°C and DNA was extracted using the ChIP DNA Clean and Concentrator system (Zymo Research). We treated 1% of the input in parallel. Libraries were prepared according to the Illumina protocol and sequenced using an Illumina NextSeq Sequencing System.

## Whole-mount staining, immunohistochemistry, and in situ hybridization

Immunohistochemistry was performed on 15 µm cryostat sections as described (*Chen et al., 2011*). Primary antibodies used in this study are detailed in the resource table. Whole-mount antibody staining was performed as described (*Dasen et al., 2008*), and GFP-labeled motor axons were visualized in projections of a Zeiss Lightsheet Z.1 microscope (400–600 µm). Unless indicated otherwise, immunohistological data shown in figures are representative of n>3 analyzed mutants. Images for control animals are from age-matched littermates. In situ hybridizations were performed as described previously (*Chen et al., 2007*; *Chen et al., 2011*) and in the Materials and Methods.

## Subcellular RNA fractionation

We followed a previously published protocol to extract subcellular fractions of RNA (*Gagnon et al., 2014*). We used TRIzol (Thermo Fisher Scientific) to extract RNA and perform reverse transcription (RT) with hexamer primers. *Gapdh* (mRNA in cytoplasm), *Rnu1* (snRNA in nucleus), and *Kcnq1ot1* (a known chromatin-associated lncRNA) were used as quality controls to verify fractionation.

## RNA pull-down assay

In vitro-transcribed biotin-labelled RNAs were generated by the Biotin RNA Labeling Mix (Roche) and T7 RNA polymerase (Promega). Templates were treated with RNase-free DNase I (Promega) and the reaction mix was purified with Oligo Clean and Concentrator (D4060, Zymo Research). Biotinylated RNA (3 μg) was heated to 65°C for 10 min and then slowly cooled down to 4°C. After that, RNA structure buffer (10 mM Tris pH 7, 0.1 M KCl, 10 mM MgCl$_2$) was added and the mix was shifted to room temperature for 20 min to allow proper secondary structure formation. Folded RNA was then mixed with 1 mg of ESC protein nuclear extract in RIP buffer (500 mM NaCl, 10 mM HEPES pH 7.5, 25% glycerol, 1 mM EDTA, and protease inhibitor) and incubated at 4°C for one hour. Twenty μL Dynabeads M-280 Streptavidin (Invitrogen) were added to each binding reaction and further incubated at room temperature for one hour. Beads were washed briefly five times and boiled in SDS buffer, and the retrieved protein was detected by standard Western blot analysis.

## Co-immunoprecipitation (Co-IP) and Western blot

For each IP, cells were harvested from a 10 cm dish and washed twice with ice cold PBS. Cell pellets were allowed to swell in twice the volume of cytoplasmic lysis buffer (50 mM NaCl, 10 mM HEPES-pH 7.5, 500 mM sucrose, 1 mM EDTA and protease inhibitors). Samples were incubated on ice for 10 min, followed by centrifugation at 2,000 rpm for 10 min. The cloudy supernatant cytoplasmic fraction was removed. After washing twice (50 mM NaCl, 10 mM HEPES-pH 7.5, 25% glycerol, 1 mM EDTA and protease inhibitors), the cell pellets were resuspended in the same volume of high salt buffer (500 mM NaCl, 10 mM HEPES-pH 7.5, 25% glycerol, 1 mM EDTA and protease inhibitors), and rotated for 30~60 min at 4°C. Then cell pellets were centrifuged at 14,000 rpm for 10 min at 4°C. The supernatant was incubated overnight at 4°C with antibody and pre-cleared Protein-G beads (depending upon the antibody) to immunoprecipitate endogenous protein against the specific antibody used. We collected 10% of cleared supernatant as input. Subsequently, IP-protein beads were washed three times with PBS and 0.01% Tween-20, each for 5 min at 4°C. IP-proteins and their interacting partners were eluted from beads in 6X reducing loading buffer at 70°C for 15 min. Finally, samples were cooled down to room temperature and spun briefly to collect condensation. Standard Western blot procedures were applied using anti-Jarid2 (Novus Biologicals, NB100-2214) or anti-Ezh2 (Millipore, 17–662) antibodies. Blots were developed using HRP-conjugated anti-rabbit or -mouse antibodies, depending on the species of the primary antibody. Signals were developed and filmed by enhanced SuperSignal West Femto Maximum Sensitivity Substrate (Thermo, 34096). All exposures were done using hyper film.

## Single molecular RNA FISH

ESC~MNs were cultured and harvested on slides. Cells were fixed in 4% paraformaldehyde for 10 min at room temperature, permeabilized for 5 min on ice in PBS with 0.5% Triton X-100, and then rinsed in 70% EtOH for subsequent RNA FISH. Slides and coverslips were kept in 70% EtOH at 4°C until staining. Slides were then washed in wash buffer (10% deionized formamide in 2X SSC) for 5 min and incubated in a dark room at 37°C for at least 4 hr with 1 μL of probe stock solution and 100 μL of hybridization buffer (1 g dextran sulfate, 1 mL 20X SSC, 1 mL deionized formamide). *Meg3* smFISH probes were purchased from Stellaris. Images were captured with a Delta Vision microscopy system.

## RNA immunoprecipitation (RIP)

RIP was performed with the RNA-Binding Protein Immunoprecipitation Kit (17–700, Millipore) according to the manufacturer's protocol with some modifications. ESC~MNs were dissociated at a concentration of 2 million cells/mL and treated with 0.3% formaldehyde in ice-cold PBS for 10 min at 37°C. Glycine/PBS was added to a final concentration of 0.125 M and each sample was incubated for 5 min at room temperature. After crosslinking, ten million cells were washed twice with cold PBS and then suspended in 100 μL RIP lysis buffer (with the addition of protease inhibitor and RNase inhibitor). The lysate was incubated on ice for 5 min and centrifuged at 14,000 rpm for 10 min at 4°C. Ezh2, Jarid2, and Suz12 antibodies were added for respective IP reactions and then incubated in RIP buffer (0.5 M EDTA/RNase inhibitor) for 3 hr to overnight at 4°C. Samples were washed at least five times with RIP washing buffer. RIP beads were resuspended in RIPA buffer (RIP washing

buffer +10% SDS +protease K) to reverse crosslinking at 56°C for 30 min. RNA samples were extracted and qPCR was performed as described above. Isolated proteins before proteinase K treatment were collected from the beads and verified by Western blot analysis. Data on retrieved RNAs was calculated from the RT/input ratio for each experiment.

## Statistical analyses and graphical representations

All statistical analyses were generated with GraphPad Prism 6 (GraphPad Software). The values are shown as mean ± SD, as indicated. Student's *t*-tests were used for comparisons between experimental samples and controls. Statistical significance was defined as * $p<0.05$ and ** $p<0.01$ by Student's *t*-test.

## RNA-seq analysis

Adapter contamination in the paired-end reads was removed using PEAT (*Li et al., 2015*), and the trimmed reads were aligned to the mm10 genome with STAR (*Dobin et al., 2013*). The standard GTF-formatted transcript annotation was defined by GENCODE (version M9) (*Harrow et al., 2012*), which includes many evidence-based lncRNAs. We used this annotation to aid the junction read alignment in STAR, the output of which was submitted to Cufflinks (*Trapnell et al., 2010*) for de novo transcript assembly with the option 'library type; first-strand' to allow strand-specific alignments. We followed a strategy for novel lncRNA identification similar to that suggested by a previous report (*Qian et al., 2016*), by which only transcripts that were longer than 200 bp, had no overlap with any known genes, and consisted of more than one exon were regarded as novel lncRNAs. We pooled these novel lncRNAs along with all known genes annotated in GENCODE and used HTseq (*Anders et al., 2015*) to calculate the read count aligned onto each transcript. This procedure was repeated for all RNA-seq samples in this study. The read counts of all transcripts among different samples were normalized using a TMM algorithm with the trimming option M = 30% and A = 5% (*Robinson and Oshlack, 2010*). A general comparison of different normalization algorithms can be found in *Lin et al. (2016)*. We calculated the specificity score of each transcript among the samples at different stages according to the Jensen–Shannon definition for tissue specificity scores (*Cabili et al., 2011*; *Trapnell et al., 2010*). The transcripts were split into three groups—namely protein coding genes, annotated lncRNAs, and novel lncRNAs—for which specificity score distributions were plotted and compared.

## ChIP-seq analysis

Reads were trimmed by PEAT and aligned to the mm10 genome using Bowtie2 (*Langmead and Salzberg, 2012*). Following a similar flow analysis described in our previous work (*Chen et al., 2013*; *Yildirim et al., 2011*), all alignments were extended downstream to span an exact 150 bp-long region. Extensions that exceeded the ends of chromosomes were clipped. The extended alignments were input into the *genomecov* functionality supported in the BEDTools suite (*Quinlan and Hall, 2010*) to generate read coverage profiles at a base-pair resolution. The coverage for each chromosomal position was normalized according to the mappable read count. Each sample was averaged and binned to reveal major trends. To identify possible differentially-enriched histone marks among stages or treatments, we used MACS 1.4 (*Feng et al., 2011*) to call peaks (p-value<$10^{-5}$) in each ChIP-seq sample with the corresponding input library and then overlapping peaks were merged using MAnorm (*Shao et al., 2012*) to reveal loci with a significant change between two samples.

## Axon arborization quantification with Imaris

The 3D images acquired with a Zeiss Lightsheet Z.1 microscope were subjected to analyses in Imaris 8.4.0 (Bitplane, Zurich, Switzerland) for quantification of axon arborization. Regions of interest were segmented for detection of individual neurons. Motor nerve terminals were semi-automatically traced using the filament tracer wizard from a defined starting point. The AutoPath (no loops) algorithm was selected. Seed points detected from background signals were manually removed. Disconnected segments were removed by indicating the maximum gap length, and background subtraction was applied for noise removal. The 'Filament No. Dendrite Terminal Points' tool automatically calculated the number of motor nerve terminals.

## Acknowledgements

We thank people in the Chen lab, particularly to Hung Lo, for reading and giving critical comments to this manuscript. Dr. Mei-Yeh Lu from the NGS Core in Academia Sinica for invaluable technical advice and help in performing RNA-seq. Experiments and data analysis were performed in part through the use of the advanced optical microscopes at the Division of Instrument Services of Academia Sinica and with the assistance of Shu-Chen Shen. We appreciate the Genomic, FACS and Imaging core facilities in IMB for considerable technical assistance. The *IG-DMR*<sup>matΔ</sup> line was a kind gift from Prof. Ann Fergusson Smith of the University of Cambridge, UK. The NIL plasmid was a gift from Prof. Hynek Wichterle from Columbia University, USA. We also acknowledge Bernhard Payer (CRG, Spain) and Wee-Wei Tee (A*STAR, Singapore) for their insightful and critical comments and for discussing the experimental results. The IMB's Scientific English Editing Core reviewed the manuscript. This work is funded by Academia Sinica Career Development Award (AS-CDA-107-L05), MoST (104 – 2311-B-001 – 030-MY3), and NHRI (NHRI-EX106-10315NC). SPL was supported by MoST (104 – 2321-B-002 – 043) and JHH was supported by MoST (104 – 2311-B-009 – 002-MY3 and 105 – 2221-E-009 – 126-MY3).

## Additional information

### Funding

| Funder | Grant reference number | Author |
| --- | --- | --- |
| Ministry of Science and Technology | 104 – 2321-B-002 – 043 | Shau-Ping Lin |
| Ministry of Science and Technology | 104 – 2311-B-009 – 002-MY3 | Jui-Hung Hung |
| Ministry of Science and Technology | 105 – 2221-E-009 – 126-MY3 | Jui-Hung Hung |
| National Health Research Institutes | NHRI-EX106-10315N | Jun-An Chen |
| Ministry of Science and Technology | 104-2311-B-001 -030 -MY3 | Jun-An Chen |
| Academia Sinica | AS-CDA-107-L05 | Jun-An Chen |

The funders had no role in study design, data collection and interpretation, or the decision to submit the work for publication.

### Author contributions

Ya-Ping Yen, Resources, Data curation, Software, Formal analysis, Validation, Investigation, Visualization, Methodology, Writing—original draft, Writing—review and editing; Wen-Fu Hsieh, Joye Li, Software; Ya-Yin Tsai, Validation, Investigation, Writing—review and editing; Ya-Lin Lu, Formal analysis; Ee Shan Liau, Ho-Chiang Hsu, Yen-Chung Chen, Ting-Chun Liu, Validation; Mien Chang, Experimental mice breeding; Shau-Ping Lin, Jui-Hung Hung, Supervision; Jun-An Chen, Conceptualization, Resources, Data curation, Formal analysis, Supervision, Funding acquisition, Validation, Investigation, Methodology, Writing—original draft, Project administration, Writing—review and editing

### Author ORCIDs

Ya-Ping Yen http://orcid.org/0000-0003-2264-8126
Ya-Yin Tsai http://orcid.org/0000-0002-9152-7646
Ee Shan Liau http://orcid.org/0000-0002-4115-5573
Yen-Chung Chen http://orcid.org/0000-0002-8529-1251
Jui-Hung Hung https://orcid.org/0000-0003-2208-9213
Jun-An Chen http://orcid.org/0000-0001-9870-3203

### Ethics

Animal experimentation: All of the live animals were kept in an SPF animal facility, approved and overseen by IACUC (12-07-389 ) Academia Sinica.

### Decision letter and Author response

Decision letter https://doi.org/10.7554/eLife.38080.sa1
Author response https://doi.org/10.7554/eLife.38080.sa2

## Additional files

### Supplementary files

• Supplementary file 1. Cell type enriched lncRNAs from ESC to ESC~MNs. Values indicate the stage-specific scores (see Materials and methods), 70 stage-signature lncRNAs during the ESC~MN differentiation process are uncovered (ESC, NE, pMN, MN, and IN showed in *Figure 1C*).

• Supplementary file 2. MN-enriched lncRNAs uncovered from RNA-seq. Only the highly expressed (TMM normalized read count >10 in all samples) lncRNAs are shown.

• Transparent reporting form

### Data availability

All microarray, RNA-seq, ChIP-seq data have been deposited in GEO under accession codes GSE114283, GSE114285 and GSE114228.

The following datasets were generated:

| Author(s) | Year | Dataset title | Dataset URL | Database and Identifier |
|---|---|---|---|---|
| Jun-An Chen, Ya-Ping Yen | 2018 | Genome-wide maps of H3K27me3 in chromatin state in embryonic stem cells differentiated motor neurons | http://www.ncbi.nlm.nih.gov/geo/query/acc.cgi?acc=GSE114283 | NCBI Gene Expression Omnibus, GSE114283 |
| Jun-An Chen, Ya-Ping Yen | 2018 | Transcriptome analysis of Meg3 KD and IG-DMR maternal deletion in ESC, pMN, and MN | http://www.ncbi.nlm.nih.gov/geo/query/acc.cgi?acc=GSE114228 | NCBI Gene Expression Omnibus, GSE114228 |
| Jun-An Chen, Ya-Ping Yen | 2018 | Next Generation Sequencing Facilitates Quantitative Analysis of ES, pMN, MN, and IN Transcriptomes | http://www.ncbi.nlm.nih.gov/geo/query/acc.cgi?acc=GSE114285 | NCBI Gene Expression Omnibus, GSE114285 |

The following previously published datasets were used:

| Author(s) | Year | Dataset title | Dataset URL | Database and Identifier |
|---|---|---|---|---|
| Mazzoni EO, Mahony S, Morrison CA, Nedelec S, Gifford DK, Wichterle H | 2013 | Induced V5-tagged Lhx3 (iLhx3-V5) in iNIL3-induced motor neurons (Day 4) | https://www.ncbi.nlm.nih.gov/geo/query/acc.cgi?acc=GSM782847 | NCBI Gene Expression Omnibus, GSM782847 |
| Mazzoni EO, Mahony S, Morrison CA, Nedelec S, Gifford DK, Wichterle H | 2013 | Isl1/2 in iNIL3-induced motor neurons (Day 4) | http://www.ncbi.nlm.nih.gov/geo/query/acc.cgi?acc=GSM782848 | NCBI Gene Expression Omnibus, GSM782848 |
| Narendra V, An D, Mazzoni E, Reinberg D | 2015 | H3K4me3 | http://www.ncbi.nlm.nih.gov/geo/query/acc.cgi?acc=GSM1468401 | NCBI Gene Expression Omnibus, GSM1468401 |
| Rhee H, Closser M, Guo Y, Bashkirova EV, Tan GC, Gifford DK, Wichterle H | 2016 | H3K27ac_day6 | http://www.ncbi.nlm.nih.gov/geo/query/acc.cgi?acc=GSM2098385 | NCBI Gene Expression Omnibus, GSM2098385 |
| Rhee H, Closser M, Guo Y, Bashkirova | 2016 | ATAC_seq_day6 | http://www.ncbi.nlm.nih.gov/geo/query/acc.cgi? | NCBI Gene Expression Omnibus, |

| | | | | | |
|---|---|---|---|---|---|
| EV, Tan GC, Gifford DK, Wichterle H | | | | acc=GSM2098391 | GSM2098391 |
| Mazzoni EO, Mahony SA, McCuine S, Young RA, Wichterle H, Gifford DK | 2011 | RAR_Day2+8hrsRA | | http://www.ncbi.nlm.nih.gov/geo/query/acc.cgi?acc=GSM482750 | NCBI Gene Expression Omnibus, GSM482750 |
| Mazzoni EO, Mahony SA, McCuine S, Young RA, Wichterle H, Gifford DK | 2011 | Pol2-S5P_Day2+8h | | http://www.ncbi.nlm.nih.gov/geo/query/acc.cgi?acc=GSM981593 | NCBI Gene Expression Omnibus, GSM981593 |
| Chen J, Yen Y | 2017 | ES-WT | | https://www.ncbi.nlm.nih.gov/geo/query/acc.cgi?acc=GSM2420680 | NCBI Gene Expression Omnibus, GSM2420680 |
| Chen J, Yen Y | 2017 | AK4-WT | | https://www.ncbi.nlm.nih.gov/geo/query/acc.cgi?acc=GSM2420683 | NCBI Gene Expression Omnibus, GSM2420683 |
| Chen J, Yen Y | 2017 | AK7-WT | | https://www.ncbi.nlm.nih.gov/geo/query/acc.cgi?acc=GSM2420684 | NCBI Gene Expression Omnibus, GSM2420684 |

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
