## [Decision Letter]

Thank you for sending your article entitled "*Dlk1-Dio3* Locus-Derived LncRNAs Perpetuate Postmitotic Motor Neuron Cell Fate and Subtype Identity" for peer review at *eLife*. Your article is being evaluated by two peer reviewers, and the evaluation is being overseen by a Reviewing Editor and Marianne Bronner as the Senior Editor.

Summary:

The function of most lncRNAs in nervous system development remains largely unknown. This study by Yen et al. aims to address this knowledge gap by focusing on the imprinted *Dlk1-Dio3* locus, which produces three protein-coding genes (*Dlk1, Rtl1, Dio3*) from the paternally inherited allele and multiple lncRNAs (*Meg3, Rian, Mirg*) and small ncRNAs from the maternally inherited allele. Although this locus has been studied for years, the authors have discovered a new role for the *Dlk1-Dio3* locus-derived lncRNAs in post-mitotic motor neurons (MNs). The authors find that *Meg3* is induced in MN differentiation in response to retinoic acid and that *Meg3* affects PRC2-Jarid2 complex assembly. *Meg3* knockdown cause H3K27me3 depletion and gene induction at several 5' *Hox* genes and the authors nicely show that this is associated with altered motor neuron identity and nerve targeting in vivo. Through a combination of elegant biochemical approaches in mESC-derived MNs, the authors find that these lncRNAs function through PRC2 (via gene repression) and have a dual role: (a) repression of progenitor/non-neural fate, and (b) establishment of MN subtype identity via repression of caudal *Hox* genes. This work represents a significant contribution as it aims to uncover the role of the lncRNAs of the mammalian imprinted *Dlk1-Dio3* locus in neuronal development. In addition, the authors devote significant effort into translating the in vitro mESC-derived MN findings to the mouse spinal cord. That being said, the authors should address the following concerns.

Essential revisions:

1) A major conclusion of this paper is that *Dlk1-Dio3* locus-derived lncRNAs (*Meg3*) repress progenitor genes in MNs. However, it has not been demonstrated that *Dbx1* de-repression occurs in post-mitotic MNs of IG DMR mice. This point could be further consolidated by discovering additional progenitor genes that are de-repressed in vivo in IG DMR mice. The list generated via RNA-seq on mESC-derived MNs contains additional candidates (*Pax6, Neurog2*).

2) By performing *Meg3* knock-down in mESC-derived MNs, the authors show effects on progenitor genes and *Hox* genes. However, it remains to be shown in vivo whether loss or knock-down of *Meg3* results in similar effects. They indirectly conclude that this is the case by using the IG DMR mice (although *Meg3* down-regulation in post-mitotic MNs of IG DMR mice is not shown), but in these mice concomitant up-regulation of paternal coding genes occurs (Lin et al., 2003, 2007), including *Dlk1* (known to affect MN development). In addition, the IG DMR mESC-MNs show massive differences in gene expression that are not found in *Meg3* KD ESC-MNs (Figure 4C). These differences could contribute to the phenotypes observed in IG DMR spinal cords. This concern could be addressed by a more targeted approach to knock-out or knock-down (shRNA) *Meg3* in the mouse spinal cord. Along the same lines, I do not understand the argument made in Discussion (subsection “Combinatorial and individual roles of the *Meg3-Rian-Mirg* lncRNA cluster”) about not using the available mouse KO lines for *Meg3*. The *Meg3* KO mice represent a more elegant/targeted approach to perturb *Meg3* compared to IG DMR mice.

3) Through elegant biochemistry, the authors show that *Meg3* interacts with PRC2 components and Jarid2. If this model is correct, we can make 2 predictions: (a) ChIP for PRC2 (Ezh2, Suz12) or Jarid2 and subsequent qPCR will reveal binding to progenitor genes and *Hox* genes. (b) And if *Meg3* is required for Ezh2/Jarid2 interaction, then PRC2/Jarid2 binding to DNA should be compromised in *Meg3* KD ESC-derived MNs. The authors have the necessary expertise for this experiment.

4) Likewise, *Meg3* interaction with Ezh2 and Jarid2: The authors find that genetic manipulation of *Meg3*(V5) isoform affects the co-IP of Ezh2 and Jarid2, and conclude that *Meg3* facilitates Ezh2-Jarid2 complex assembly directly. This is a major claim of the paper that is not fully substantiated. What is the copy number of Ezh2, Jarid2, and *Meg3*? Is the copy number of *Meg3* sufficient to mediate this effect? Are there independent binding sites on *Meg3* for different subunits of the PRC2-Jarid complex, and how can the authors rule out indirect effects of *Meg3* on cell state that affects PRC2 complex assembly?

5) The authors use biotinylated *Meg3* RNA to retrieve Ezh2 from MN cell extract, and then conclude this demonstrates "direct interaction" between *Meg3* and Ezh2 (subsection “*Meg3* facilitates interaction of the PRC2 complex with Jarid2 in MNs”, first paragraph). This is incorrect as the extract contains a multitude of proteins. UV crosslinking in cells or in vitro reconstitution with purified components is needed to make such a conclusion.

6) The presence of multiple long and small regulatory RNAs from the *Dlk1-Dio3* locus somewhat complicates the interpretation of these results. The use of *Meg3*(v5) transgene to rescue is informative, but this was only done for a subset of experiments. The authors should take this into account more in the Discussion.

Major concerns that do not require experiments:

1) Why were the experiments shown in Figure 3C-G done on ESC and not ESC-derived MNs? At minimal, an explanation must be provided. This is important because at the end of the subsection “*Meg*3 facilitates interaction of the PRC2 complex with Jarid2 in MNs”, the authors conclude that *Meg3* facilitates binding of PRC2 and Jarid2 in post-mitotic MNs. Also, this argument could be strengthened by providing evidence that Jarid2 is expressed in vivo in spinal MNs. Similarly, it appears that *Meg3* knock-down was performed at the ESC stage, not at the progenitor MN or post-mitotic MN stage? The reason for this should be clearly stated in Results because this point is raised in Discussion (subsection “PRC1/2 with lncRNA: a fail-safe mechanism to guard MN epigenetic landscape”, last paragraph) but with little justification.

2) The authors compare data from other studies performed in ESC-derived MNs. However, it is not clearly stated whether all these studies generate the same "type" of MNs. Do all these previous studies generate mESC-derived MNs of "rostral" identity? If yes, this should be added in the text because: (i) it makes sense then that by profiling these MNs the authors found *Meg3*, which is highly expressed in rostral spinal cord, (ii) it will also help the reader in Discussion (subsection “PRC2-lncRNA regulation in MNs”, first paragraph) when the authors talk about cervical MNs.

3) In Discussion, similarities and differences between the mechanism described by the authors and previous studies on the lncRNA *HOTAIR* (also works with PRC2 to repress caudal *Hox*) could be discussed (PMID: 17604720, PMID: 20616235).

4) The authors find that *Meg3* is enriched in rostral MNs. However, in Figure 4—figure supplement 1E, *Meg-3* knock-down results in loss of rostral *Hox* gene expression and gain of caudal *Hox* gene expression. The authors should at least discuss how Polycom-mediated gene repression results in these 2 different effects on *Hox* gene expression, especially in light of the fact that these results are not phenocopied in vivo with the IG-DMR mice.

Reviewer #1:

Yen et al. present an integrated analysis of the *Meg3* imprinted lncRNA in motor neuron differentiation. The authors find that *Meg3* is induced in MN differentiation in response to retinoic acid, and that *Meg3* affects PRC2-Jarid2 complex assembly. *Meg3* knock down cause H3K27me3 depletion and gene induction at several 5' *Hox* genes, and the authors nicely show that this is associated with altered motor neuron identity and nerve targeting in vivo. The strength of this work is the use of rigorous genetic models to validate the model that lncRNA regulates *Hox* loci in trans and thereby alter neuronal cell positional identity. The mechanistic details are not yet fully clear, which the authors should be more cautious in their interpretations.

1) *Meg3* interaction with Ezh2 and Jarid2: The authors find that genetic manipulation of *Meg3*(V5) isoform affects the co-IP of Ezh2 and Jarid2, and conclude that *Meg3* facilitates Ezh2-Jarid2 complex assembly directly. This is a major claim of the paper that is not fully substantiated. What is the copy number of Ezh2, Jarid2, and *Meg3*? Is the copy number of *Meg3* sufficient to mediate this effect? Are there independent binding sites on Meg3 for different subunits of the PRC2-Jarid complex, and how can the authors rule out indirect effects of *Meg3* on cell state that affects PRC2 complex assembly?

2) The authors use biotinylated *Meg3* RNA to retrieve Ezh2 from MN cell extract, and then conclude this demonstrates "direct interaction" between *Meg3* and Ezh2 (subsection “*Meg3* facilitates interaction of the PRC2 complex with Jarid2 in MNs”, first paragraph). This is incorrect as the extract contains a multitude of proteins. UV crosslinking in cells or in vitro reconstitution with purified components is needed to make such a conclusion.

3) The presence of multiple long and small regulatory RNAs from the *Dlk1-Dio3* locus somewhat complicates the interpretation of these results. The use of *Meg3*(v5) transgene to rescue is informative, but this was only done for a subset of experiments. The authors should take this into account more in the Discussion.

Reviewer #2:

The function of most lncRNAs in nervous system development remains largely unknown. This study by Yen et al. aims to address this knowledge gap by focusing on the imprinted *Dlk1-Dio3* locus, which produces three protein coding genes (*Dlk1, Rtl1, Dio3*) from the paternally inherited allele and multiple lncRNAs (*Meg3, Rian, Mirg*) and small ncRNAs from the maternally inherited allele. Although this locus has been studied for years, the authors have discovered a new role for the *Dlk1-Dio3* locus-derived lncRNAs in post-mitotic motor neurons (MNs). By using a well-established in vitro differentiation protocol to generate MNs from stem cells, the authors first provide a mini-resource (although the majority of genes were not validated) of lncRNAs that show cell-type specificity during different stages of MN development (NE, pMN, MN). This resource revealed that the *Dlk1-Dio3* locus-derived lncRNAs (*Meg3, Rian, Mirg*) are enriched in post-mitotic MNs in vitro, and further demonstrated that *Meg3* is expressed in developing mouse spinal MNs in vivo (high rostral/low caudal). Through a combination of elegant biochemical approaches in mESC-derived MNs, the authors find that these lncRNAs function through PRC2 (via gene repression) and have a dual role: (a) repression of progenitor/non-neural fate, and (b) establishment of MN subtype identity via repression of caudal *Hox* genes.

This work represents a significant contribution as it aims to uncover the role of the lncRNAs of the mammalian imprinted *Dlk1-Dio3* locus in neuronal development. In addition, the authors devote significant effort into translating their in vitro mESC-derived MN findings to the mouse spinal cord. However, I do have several major concerns that moderated enthusiasm.

Major concerns that require experiments:

1) A major conclusion of this paper is that *Dlk1-Dio3* locus-derived lncRNAs (*Meg3*) repress progenitor genes in MNs. However, it has not been demonstrated that *Dbx1* de-repression occurs in post-mitotic MNs of IG DMR mice. This point could be further consolidated by discovering additional progenitor genes that are de-repressed in vivo in IG DMR mice. The list generated via RNA-seq on mESC-derived MNs contains additional candidates (*Pax6, Neurog2*).

2) By performing *Meg3* knock-down in mESC-derived MNs, the authors show effects on progenitor genes and *Hox* genes. However, it remains to be shown in vivo whether loss or knock-down of *Meg3* results in similar effects. They indirectly conclude that this is the case by using the IG DMR mice (although *Meg3* down-regulation in post-mitotic MNs of IG DMR mice is not shown), but in these mice concomitant up-regulation of paternal coding genes occurs (Lin et al., 2003, 2007), including *Dlk1* (known to affect MN development). In addition, the IG DMR mESC-MNs show massive differences in gene expression that are not found in *Meg3* KD ESC-MNs (Figure 4C). These differences could contribute to the phenotypes observed in IG DMR spinal cords. This concern could be addressed by a more targeted approach to knock-out or knock-down (shRNA) *Meg3* in the mouse spinal cord. Along the same lines, I do not understand the argument made in Discussion (subsection “Combinatorial and individual roles of the *Meg3-Rian-Mirg* lncRNA cluster”) about not using the available mouse KO lines for *Meg3*. The *Meg3* KO mice represent a more elegant/targeted approach to perturb Meg3 compared to IG DMR mice.

3) Through elegant biochemistry, the authors show that *Meg3* interacts with PRC2 components and Jarid2. If this model is correct, we can make 2 predictions: (a) ChIP for PRC2 (Ezh2, Suz12) or Jarid2 and subsequent qPCR will reveal binding to progenitor genes and *Hox* genes. (b) And if *Meg3* is required for Ezh2/Jarid2 interaction, then PRC2/Jarid2 binding to DNA should be compromised in *Meg3* KD ESC-derived MNs. The authors have the necessary expertise for this experiment.

Major concerns that do not require experiments:

1). Why the experiments shown in Figure 3C-G were done on ESC and not ESC-derived MNs? At minimal, an explanation must be provided. This is important because at the end of the subsection “*Meg3* facilitates interaction of the PRC2 complex with Jarid2 in MNs”, the authors conclude that *Meg3* facilitates binding of PRC2 and Jarid2 in post-mitotic MNs. Also, this argument could be strengthened by providing evidence that Jarid2 is expressed in vivo in spinal MNs. Similarly, it appears that *Meg3* knock-down was performed at the ESC stage, not at the progenitor MN or post-mitotic MN stage? The reason for this should be clearly stated in Results because this point is raised in Discussion (subsection “PRC1/2 with lncRNA: a fail-safe mechanism to guard MN epigenetic landscape”, last paragraph) but with little justification.

2) The authors compare data from other studies performed in ESC-derived MNs. However, it is not clearly stated whether all these studies generate the same "type" of MNs. Do all these previous studies generate mESC-derived MNs of "rostral" identity? If yes, this should be added in the text because: (i) it makes sense then that by profiling these MNs the authors found *Meg3*, which is highly expressed in rostral spinal cord, (ii) it will also help the reader in Discussion (subsection “PRC2-lncRNA regulation in MNs”, first paragraph) when the authors talk about cervical MNs.

3) In Discussion, similarities and differences between the mechanism described by the authors and previous studies on the lncRNA *HOTAIR* (also works with PRC2 to repress caudal *Hox*) could be discussed (PMID: 17604720, PMID: 20616235).

4) The authors find that *Meg3* is enriched in rostral MNs. However, in Figure 4—figure supplement 1E, *Meg-3* knock-down results in loss of rostral *Hox* gene expression and gain of caudal *Hox* gene expression. The authors should at least discuss how Polycom-mediated gene repression results in these 2 different effects on *Hox* gene expression, especially in light of the fact that these results are not phenocopied in vivo with the IG-DMR mice.

---

## [Author Response]

Essential revisions:1) A major conclusion of this paper is that Dlk1-Dio3 locus-derived lncRNAs (Meg3) repress progenitor genes in MNs. However, it has not been demonstrated that Dbx1 de-repression occurs in post-mitotic MNs of IG DMR mice. This point could be further consolidated by discovering additional progenitor genes that are de-repressed in vivo in IG DMR mice. The list generated via RNA-seq on mESC-derived MNs contains additional candidates (Pax6, Neurog2).

We thank the reviewer for this suggestion to strengthen our conclusion that progenitor genes are derepressed in the *IG-DMR^matΔ^* embryos. In revised new Figure 5—figure supplement 1A, we first verified that the expression of *Meg3* is still lacking in the developing spinal cord of E14.5 *IGDMR^matΔ^* embryos. We then checked the progenitor proteins Pax6, Irx3and Dbx1 (data not shown for Dbx1 as Dbx1 antibody did not work well for immunostaining). Compared to the control littermates, we observed a significant increase in the percentage of Pax6^on^ (45% penetrance, n = 5/11), and Irx3^on^ cells (100% penetrance, n = 8/8) for postmitotic MNs (Isl1/2^on^) along the entire rostrocaudal axis of the dorsal spinal cord in the *IG-DMR^matΔ^* embryos (new Figures 5A and 5C, only the cervical segment is shown; quantifications shown in new Figures 5B and 5D). However, Hb9^on^ and Isl1(2)^on^ MNs were comparable between the control and *IG-DMR^matΔ^* embryos at E10.5 (new Figures 5E and 5F). Although dorsal progenitor genes were aberrantly up-regulated in the postmitotic MNs, production of MNs remained relatively unaffected, suggesting that the generation of MNs is still intact with the co-expressed progenitor/postmitotic genes in the *IG-DMR^matΔ^* embryos (new Figure 5G). This outcome is consistent with the postmitotic expression of *Meg3* and its function to maintain the silenced epigenetic state of progenitor genes.

Although we observed a complete 100% penetrance of Hoxa5-c8 cell fate switch in the rostrobrachial segments and an increase co-expression of Irx3^on^Hb9^on^ cells in the *IG-DMR^matΔ^* embryos (Figures 5 and 6), ectopic Pax6 was manifested at partial penetrance (45% , 5 in the 11 mutants) and several caudal Hox protein (including Hox9 and 10) were not shown to display significant changein vivo (Figures 5 and Figure 5—figure supplement 1), consistent with the finding that removal of *Ezh2* from MN progenitors has no detectable impact on segmental Hox expression in the spinal cord (Hoxc6 and Hoxc9) (Golden and Dasen, 2012). This result also suggests that a compensatory PRC1-mediated function in vivo might make up for the loss of *Meg3*-mediated epigenetic maintenance in segmental MNs (Golden and Dasen, 2012; Mazzoni et al., 2013). Our results are not entirely unexpected as many potent miRNA/lncRNA KO mice reflect either only partial phenotype penetrance (Li et al., 2013; Li et al., 2016; Medeiros et al., 2011), or more severe phenotypes upon genetic or environmental stresses (Williams et al., 2009). Moreover, many miRNAs/TFs function as repressors to silence progenitor/neighboring interneuron genes (Chen et al., 2011; Shirasaki and Pfaff, 2002), they may also constitute a coherent loop with lncRNAs to safeguard the terminal postmitotic cell fate with a fail-safe control. This part is now added in the Discussion subsection “A fail-safe mechanism to guard MN epigenetic landscape”, last paragraph.

2) By performing Meg3 knock-down in mESC-derived MNs, the authors show effects on progenitor genes and Hox genes. However, it remains to be shown in vivo whether loss or knock-down of Meg3 results in similar effects. They indirectly conclude that this is the case by using the IG DMR mice (although Meg3 down-regulation in post-mitotic MNs of IG DMR mice is not shown), but in these mice concomitant up-regulation of paternal coding genes occurs (Lin et al., 2003, 2007), including Dlk1 (known to affect MN development). In addition, the IG DMR mESC-MNs show massive differences in gene expression that are not found in Meg3 KD ESC-MNs (Figure 4C). These differences could contribute to the phenotypes observed in IG DMR spinal cords. This concern could be addressed by a more targeted approach to knock-out or knock-down (shRNA) Meg3 in the mouse spinal cord. Along the same lines, I do not understand the argument made in Discussion (subsection “Combinatorial and individual roles of the Meg3-Rian-Mirg lncRNA cluster”) about not using the available mouse KO lines for Meg3. The Meg3 KO mice represent a more elegant/targeted approach to perturb Meg3 compared to IG DMR mice.

We appreciate the reviewer’s comment and apologize for not more clearly explaining our rationale. We did not use *Meg3* KO as both *Meg3* KO mice lines manifested loss of maternal *Rian* and *Mirg* lncRNA expression.In particular, the Japanese *Meg3* KO mice in which ~10 kb is deleted, including the *Meg3*-DMR region and the first five exons of *Meg3* (Takahashi et al., 2009), also causes aberrant expression of paternal genes. This outcome is consistent with our results that most, if not all, lncRNAs in the *Meg3-Rian-Mirg* locus are reduced upon *Meg3* KD and in IG-*DMR^MatΔ^*. Therefore, it remains technically challenging to obtain a specific *Meg3* KO without abrogating the expression levels of downstream lncRNAs. We hope the reviewer can appreciate that it is a longstanding challenge in the field to dissect the role of maternal lncRNAs in the regulation of *Dlk1Dio3* locus imprinting control (McMurray and Schmidt, 2012; Steshina et al., 2006; Takahashi et al., 2009; Zhou et al., 2010), and that several labs have attempted to tackle the topic but failed to generate a clear answer. We now explain our rationale more clearly in the revised Discussion (subsection “Combinatorial and individual roles of the *Meg3-Rian-Mirg* lncRNA cluster”).

As our current and previous results indicated that the expressions of most lncRNAs in the *Meg3-Rian-Mirg* locus are reduced upon *Meg3* KD and in *IG-DMR^MatΔ^* (Figure 4—figure supplement 1A)(Lin et al., 2003), it remains puzzling if these ncRNAs work independently or synergistically in this locus to regulate MNs. To further parse this question, we generated two single lncRNA *Rian^Δ/Δ^*and *Mirg^Δ/Δ^* ESCs respectively by using CRISPR-Cas9 mediated approaches (new Figure 8A). The design of the targeted deletion regions of *Rian* and *Mirg* followed two previous studies (Han et al., 2014; Labialle et al., 2014), which led to a 23 kb deletion in the *Rian^Δ/Δ^* ESC and a 20 kb deletion in the *Mirg^Δ/Δ^* ESC (new Figure 8A). We first verified that the paternal gene (i.e., *Dlk1*) is not affected in either *Rian^Δ/Δ^*or *Mirg^Δ/Δ^* ESCs. In the *Rian^Δ/Δ^*ESCs, expressions of *Meg3* and *Mirg* were relatively unaffected, whereas that of *Rian* was compromised. Conversely, only expression of *Mirg* was impaired significantly in the *Mirg^Δ/Δ^* ESCs, but expressions of *Meg3* and *Rian* manifested minimal changes in that cell line (new Figure 8B). Upon differentiation, *Meg3* KD MNs showed ectopic up-regulated expressions of progenitor and caudal *Hox* genes (Figure 4 and new Figure 8C). In contrast, expression of *Hoxa5* remained unchanged in the *Rian^Δ/Δ^*and *Mirg^Δ/Δ^* MNs, while *Pax6* and *Hoxc8* expressions were restored to similar levels to control ESC~MNs (new Figure 8C). These results indicate that *Meg3* might be the major regulatory lncRNA responsible for the observed MN phenotype displayed by the *IG-DMR^matΔ^* embryos (new Figure 8D).

Although our new results indicate that *Meg3* might be the major regulatory lncRNA and largely accounts for the observed MN phenotype observed in the *IG-DMR^matΔ^* embryos, we cannot fully exclude the possibility of synergistic effects of the *Meg3-Rian-Mirg* locus. We discuss this possibility with new paragraphs in our Discussion section as follows:

“Combinatorial and individual roles of the *Meg3-Rian-Mirg* lncRNA cluster

There have been multiple efforts to dissect the functions of the maternally-expressed lncRNAs in the *Meg3-Rian-Mirg* locus over the past decade (McMurray and Schmidt, 2012; Steshina et al., 2006; Takahashi et al., 2009; Zhou et al., 2010), but definitive results remain elusive. […] We anticipate that a detailed map of individual and synergetic lncRNA/miRNA functions during neural development attributable to this imprinted locus will be uncovered in the near future.”

3) Through elegant biochemistry, the authors show that Meg3 interacts with PRC2 components and Jarid2. If this model is correct, we can make 2 predictions: (a) ChIP for PRC2 (Ezh2, Suz12) or Jarid2 and subsequent qPCR will reveal binding to progenitor genes and Hox genes. (b) And if Meg3 is required for Ezh2/Jarid2 interaction, then PRC2/Jarid2 binding to DNA should be compromised in Meg3 KD ESC-derived MNs. The authors have the necessary expertise for this experiment.

As previous studies have elegantly performed Ezh2/Suz12 ChIP-seq in ESC~MNs (using the same approach applied in this study), we have checked these available ChIP-seq data (Mazzoni et al., 2013; Narendra et al., 2015) and revealed that Ezh2 enrichment peaks are concordant with our H3K27me3 binding sites in the progenitor (i.e., *Pax6, Irx3, Dbx1*, and *Neurog2*) and caudal *Hox* genes. This information has been added into new Figure 4—figure supplement 2A, B.

We also verified that upon *Meg3* KD, binding of Ezh2/Jaird2 to progenitor (i.e., *Pax6*) and caudal *Hox* (i.e., *Hoxc8*) genes is concomitantly reduced (new Figure 4—figure supplement 2C-E).

4) Likewise, Meg3 interaction with Ezh2 and Jarid2: The authors find that genetic manipulation of Meg3(V5) isoform affects the co-IP of Ezh2 and Jarid2, and conclude that Meg3 facilitates Ezh2-Jarid2 complex assembly directly. This is a major claim of the paper that is not fully substantiated. What is the copy number of Ezh2, Jarid2, and Meg3? Is the copy number of Meg3 sufficient to mediate this effect? Are there independent binding sites on Meg3 for different subunits of the PRC2-Jarid complex, and how can the authors rule out indirect effects of Meg3 on cell state that affects PRC2 complex assembly?5) The authors use biotinylated Meg3 RNA to retrieve Ezh2 from MN cell extract, and then conclude this demonstrates "direct interaction" between Meg3 and Ezh2 (subsection “Meg3 facilitates interaction of the PRC2 complex with Jarid2 in MNs”, first paragraph). This is incorrect as the extract contains a multitude of proteins. UV crosslinking in cells or in vitro reconstitution with purified components is needed to make such a conclusion.

Points #4 and 5 raise three related issues:

a) Is the interaction between *Meg3* and Ezh2/Jarid2 direct?

b) Does *Meg3* bind to other proteins in the PRC2 complex via indirect interactions?

c) Is *Meg3* copy number sufficient to mediate the pervasive PRC2/Jarid2 complex?

To address these issues, we reshaped our conclusions and performed additional experiments as follows:

1) Dr. Danny Reinberg’s laboratory has shown that JARID2/EZH2 retrieved *MEG3* via PAR-CLIP, a strong indication of direct interaction (Kaneko et al., 2014a). We discuss this study more clearly in our revised manuscript (subsection “Regulatory mode of *Meg3* for the PRC2/Jarid2 complex”).

2) Previously, we performed both qPCR and single molecule FISH on *Meg3* and revealed its abundance in postmitotic MNs (Figure 2—figure supplement 1A, D). The PRC2 complex methylates histones for epigenetic silencing and associates with thousands of protein-coding and noncoding RNAs. The impact of RNA binding on PRC2 recruitment and activity were elegantly shown by previous studies (Cifuentes-Rojas et al., 2014; Davidovich et al., 2013). Based on our analysis of the RNA-seq data generated from our ESC~MN approach, we have added an analysis to our revised manuscript to show that the copy number of *Meg3/Rian* is much more abundant than that for many other *cis*- and *trans*-acting lncRNAs known to bind the PRC2 complex, including *Xist, Kcnqtlot1*, and *Hotair* (new Figure 3—figure supplement 1A).

3) We also understand that even with the above-described results, it is not possible to completely rule out that *Meg3* binds to other PRC2 proteins indirectly to contribute to PRC2/Jarid2 recruitment, activity control, and targeting mode. This topic represents a big challenge in the lncRNA/epigenetics field. Our aim in this study is to illustrate the importance and unexpected functions of lncRNAs in the well-known imprinted *Dlk1-Dio3* locus during neural development. We now highlight in our revised manuscript that future experiments, including *Meg3* ChIRP-seq/MS and Ezh2/Jarid2 CLIP-seq, will help reveal the comprehensive interaction and targeting mode of the *Meg3*/PRC2/Jarid2 interactome. Our revised discussion of this topic is as follows:

“Regulatory mode of *Meg3* for the PRC2/Jarid2 complex

The role of lncRNAs in modulating PRC2 function is well documented, but very dynamic; from recruitment, complex loading and activity control to gene targeting (Davidovich and Cech, 2015; Kretz and Meister, 2014). […] Future systematic *Meg3* ChIRP-seq analyses, together with Ezh2/Jarid2 CLIP-seq, at all stages of ESC~MN differentiation might aid in identifying the context-dependent/independent lncRNA-mediated PRC2 targeting strategy.”

6) The presence of multiple long and small regulatory RNAs from the Dlk1-Dio3 locus somewhat complicates the interpretation of these results. The use of Meg3(v5) transgene to rescue is informative, but this was only done for a subset of experiments. The authors should take this into account more in the Discussion.

In fact, unraveling the essential role of *Meg3* in the region of *Meg3-Rian-Mirg* is a longstanding question. Given that *Meg3* is more than 15 kb and has more than 10 variants, it is very challenging to address this issue. Although it is not a perfect approach, we instead generated two single lncRNA *Rian^Δ/Δ^*and *Mirg^Δ/Δ^*knockout ESC lines using CRISPR-Cas9-mediated approaches to address this issue (new Figure 8). Details are specified in point #2 above.

Major concerns that do not require experiments:1) Why were the experiments shown in Figure 3C-G done on ESC and not ESC-derived MNs? At minimal, an explanation must be provided. This is important because at the end of the subsection “Meg3 facilitates interaction of the PRC2 complex with Jarid2 in MNs”, the authors conclude that Meg3 facilitates binding of PRC2 and Jarid2 in post-mitotic MNs. Also, this argument could be strengthened by providing evidence that Jarid2 is expressed in vivo in spinal MNs. Similarly, it appears that Meg3 knock-down was performed at the ESC stage, not at the progenitor MN or post-mitotic MN stage? The reason for this should be clearly stated in Results because this point is raised in Discussion (subsection “PRC1/2 with lncRNA: a fail-safe mechanism to guard MN epigenetic landscape”, last paragraph) but with little justification.

We apologize for the confusion and typo. All of the experiments in Figure 3 were performed on ESC~MNs, rather than ESCs. We have corrected this error in the revised manuscript.

We did not knockdown *Meg3* in postmitoic MNs simply due to the technical difficulty of performing lentivirus-mediated knockdown in embryoid bodies (EBs), which are notoriously challenging to infect given their 3D spherical shape. We have therefore added the following sentence to our revised Discussion:

“Given that *Meg3* is also highly expressed in ESCs (Kaneko et al., 2014a; Mo et al., 2015), we plan to generate a targeted *Meg3* floxed allele mouse line in the future that will allow us to specifically knockout *Meg3* in MNs and recover the potential function of *Meg3* in cell-type specific contexts.”

2) The authors compare data from other studies performed in ESC-derived MNs. However, it is not clearly stated whether all these studies generate the same "type" of MNs. Do all these previous studies generate mESC-derived MNs of "rostral" identity? If yes, this should be added in the text because: (i) it makes sense then that by profiling these MNs the authors found Meg3, which is highly expressed in rostral spinal cord, (ii) it will also help the reader in Discussion (subsection “PRC2-lncRNA regulation in MNs”, first paragraph) when the authors talk about cervical MNs.

We appreciate the reviewer’s comments. We now specifically define our ESC~MNs as being cervical Hoxa5^on^ in the revised manuscript as follows:

“Consistent with our prediction, we verified that 1) the epigenetic landscapes of H3K27me3 in the progenitor and caudal *Hox* genes uncovered here are concordant with Ezh2 enrichment revealed by a previous study that used the same ESC~MN differentiation approach to generate cervical Hoxa5^on^ MNs (Figure 4—figure supplement 2A,B) (Narendra et al., 2015).”

“Together, these results strongly endorse the critical function of PRC2/lncRNA in perpetuating the postmitotic cell fate of cervical Hoxa5^on^ MNs.”

3) In Discussion, similarities and differences between the mechanism described by the authors and previous studies on the lncRNA HOTAIR (also works with PRC2 to repress caudal Hox) could be discussed (PMID: 17604720, PMID: 20616235).

We thank the reviewers for reminding us of these two landmark studies. Although these two papers were discussed and cited in our original submission, we now re-emphasize them in our revised manuscript as they inspired our study.

New revised Discussion:

“Given that lncRNAs such as *Hotair* are proposed to scaffold the PRC2 complex and guide it to specific genome loci (Rinn et al., 2007; Rinn and Chang, 2012; Tsai et al., 2010), it is tantalizing to hypothesize that *Meg3* might manifest the dual functions of scaffolding the PRC2/Jarid2 complex and guiding it to specific loci in different cell contexts. […] Generating compound lncRNA mutants that can scaffold the PRC2/Jarid2 complex will shed light on this topic.”

4) The authors find that Meg3 is enriched in rostral MNs. However, in Figure 4—figure supplement 1E, Meg-3 knock-down results in loss of rostral Hox gene expression and gain of caudal Hox gene expression. The authors should at least discuss how Polycom-mediated gene repression results in these 2 different effects on Hox gene expression, especially in light of the fact that these results are not phenocopied in vivo with the IG-DMR mice.

It is an interesting point. ESC~MNs exposed to RA activate the rostral portion of the *Hox* cluster (*Hox1*~5), while *Hox6~13* remain repressed. The transcriptional partitioning of the *Hox* cluster is mirrored at the level of chromatin. As previously described (Mazzoni et al., 2013), and also shown in this study, H3K27me3 decorates the entire *Hox* cluster in ESCs. Upon differentiation into MNs, H3K27me3 becomes restricted to the caudal segment (Narendra et al., 2015). It is still not clear how the “repressive epigenetic state of caudal Hox” is maintained in the rostral (cervical) MNs, as the PRC2 complex does not manifest a differential expression pattern along the RC axis of the spinal cord. In this study, we found that *Meg3* is enriched in the rostral (cervical) MNs, and is able to scaffold the PRC2/Jarid2 complex for the robust catalytic activity of depositing H3K27me3. We speculate that *Meg3* might be able to further help PRC2/Jarid2 to target to the caudal *Hox* loci to maintain its repressive H3K27me3 state. Upon *Meg3* KD in the cervical MNs, the silent H3K27me3 in caudal *Hox* loci is compromised. Therefore, the caudal *Hox* genes (i.e., *Hox8~13*) become ectopically up-regulated. The rostral genes are likely repressed by the well-known “posterior prevalence” effect, while the ectopically up-regulated caudal Hox proteins silence the rostral *Hox* genes directly.